**Fungi regulate response of N₂O production process to warming and**
**grazing in a Tibetan grassland**
Lei Zhong[1], Shiping Wang[2], Xingliang Xu[3], Yanfen Wang[4], Yichao Rui[5], Xiaoqi Zhou[6],
Qinhua Shen[7], Jinzhi Wang[8], Lili Jiang[2], Caiyun Luo[9], Tianbao Gu[1], Wenchao Ma[1],
Guanyi Chen[1, 10]
[1]School of Environmental Science and Engineering, Tianjin University / China-
Australia Centre for Sustainable Urban Development, Tianjin 300350, China
[2] Laboratory of Alpine Ecology and Biodiversity, Institute of Tibetan Plateau
Research, Chinese Academy of Sciences, Beijing 100101, China
[3]Key Laboratory of Ecosystem Network Observation and Modeling, Institute of
Geographic Sciences and Natural Resources, Chinese Academy of Sciences,
Beijing 100101, China
[4] University of Chinese Academy of Sciences, Beijing 100049, China
[5] Department of Soil Science, University of Wisconsin-Madison, Madison, WI 53706,
USA
[6] Tiantong National Forest Ecosystem Observation and Research Station, Center for
Global Change and Ecological Forecasting, School of Ecological and Environmental
Sciences, East China Normal University, Shanghai 200241, China
[7] Institute of Agriculture and Environment, Massey University, Private Bag 11222,
Palmerston North 4442, New Zealand.
[8] Beijing Key Laboratory of Wetland Services and Restoration, Institute of Wetland
Research, Chinese Academy of Forestry, Beijing 100091, China
[9] Key Laboratory of Adaptation and Evolution of Plateau Biota, Northwest Institute of
Plateau Biology, Chinese Academy of Sciences, Xining 810008, China
[10]School of Science, Tibet University, No. 36 Jiangsu Street, Lhasa 850012, Tibet
Autonomous Region, China
Author for correspondence:
Dr. Wenchao Ma;Prof. Guanyi Chen
School of Environmental Science and Engineering, Tianjin University / China-
Australia Centre for Sustainable Urban Development, Tianjin 300072, China
Email: mawc916@tju.edu.cn;chenguanyi@utibet.edu.cn



**Abstract**

Lack of understanding of the effects of warming and winter grazing on soil fungal contribution to nitrous oxide ($N_2O$) production process has limited our ability to predict $N_2O$ fluxes under changes in climate and land use management, because soil fungi play an important role in driving terrestrial N cycling. A controlled warming and winter grazing experiment included control (C), winter grazing (G), warming (W) and warming with winter grazing (WG) was conducted to investigate the effects of warming and winter grazing on soil $N_2O$ production potential in an alpine meadow on the Tibetan Plateau. Our results showed that soil bacteria and fungi contributed $46 \pm 2$ % and $54 \pm 2$ % to nitrification, and $37 \pm 3$ % and $63 \pm 3$ % to denitrification in control treatment, respectively. We conclude that soil fungi could be the main source for $N_2O$ production potential for the Tibetan alpine grasslands. In our results, neither warming nor winter grazing affected the activity of enzymes responsible for overall nitrification and denitrification. However, warming significantly increased the enzyme activity of bacterial nitrification and potential of $N_2O$ production from denitrification to $53 \pm 2$% and $55 \pm 3$%, respectively, but decreased them to $47 \pm 2$% and $45 \pm 3$%, respectively. Winter grazing had no such effects. Warming and winter grazing may not affect the soil $N_2O$ production potential, but climate warming can alter biotic pathways responsible for $N_2O$ production process. These findings confirm the importance of soil fungi in soil $N_2O$ production process and how its responses to environmental and land use changes in alpine meadow ecosystems. Therefore, our results provide some new insights about ecological controls on $N_2O$ production process and contribute to the development of

ecosystem nitrogen cycle model.

Keyword: warming, winter grazing, nitrification, denitrification, fungi

## 1 **Introduction**

N$_2$O emissions from soil contribute to climate warming as N$_2$O is a potent greenhouse gas (Change, 2015), it is mainly produced in soils through microbial nitrification and denitrification (Zumft, 1997). Clarifying nitrification and denitrification processes and their controlling factors will be beneficial for understanding N cycle in terrestrial ecosystems. Previous studies are mainly focused on bacterial nitrification and denitrification (Hayatsu et al., 2008; Klotz and Stein, 2008) because the conventional N cycle is thought to be controlled primarily by bacteria. However, recent studies using novel molecular techniques have shown that soil fungi are important players in terrestrial N cycling, including N$_2$O production and nitrification/denitrification in drylands or soils with high organic carbon (C) and N (Chen et al., 2015; Huang et al., 2017; Laughlin and Stevens, 2002; Marusenko et al., 2013; Zhong et al., 2018).

The Tibetan grasslands occupy approximately 40% of the Tibetan Plateau which represents 0.7-1.0% of total global N storage (Tian et al., 2006) and is required for sufficient forage production (Zheng et al., 2000). These grasslands represent one of the most vulnerable regions in the world to climate change and anthropogenic perturbation (Thompson et al., 1993;Thompson, 2000;Wang and French, 1994). A much greater than average increase in the surface temperature has been predicted to occur in this region in the future (Giorgi et al., 2001) and have profound impacts on soil N cycling in alpine grasslands. Additionally, the grasslands of the Tibetan Plateau are generally divided into two grazing seasons, i.e. summer grazing from June to September and winter

grazing from October to May (Cui et al., 2014), which host about 13.3 million domestic
yaks and 50 million sheep, with dramatically increasing numbers in future (Yao et al.,
2006). Grazing strongly affects soil N cycling, as well as plant and microbial diversity
(Hillebrand, 2008) and the stability of ecosystems (Klein et al., 2004). Previous studies
have demonstrated losses of N caused by warming (Klein et al., 2004; 2007) and that
overgrazing (Zhou et al., 2005) leads to degradation in alpine grasslands. The effects of
climate warming and grazing on the aboveground vegetation, soil physicochemical
properties, litter mass loss, bacterial communities and $N_2O$ fluxes of Tibetan alpine
grasslands have been extensively investigated (Hu et al., 2010;Li et al., 2016;Luo et al.,
2010;Rui et al., 2012;Wang et al., 2012;Zhu et al., 2015); however, most of these studies
was focused on the effect of summer grazing, little is showed the effect of winter
grazing on them (Zhu et al. 2015; Che et al. 2018). On the other hand, many studies of
Tibetan alpine grasslands are mainly focused on bacterial nitrifiers and denitrifiers or
their activities, taking these to be the key factors on $N_2O$ emission in alpine grasslands.
However, while many studies have explored N mineralization, nitrification and even
denitrification as well as bacterial nitrifiers and denitrifiers for better understanding of
$N_2O$ emission and ecosystem functioning (Yang et al., 2013;Yue et al., 2015), few
studies have been conducted to distinguish whether bacteria or fungi dominate in $N_2O$
emission and N cycling (Kato et al., 2013), especially under warming and grazing
conditions.

108       Since optimum environments for fungi and bacteria are different, they may respond

differently to environmental changes. Fungi prefer a lower temperature (Pietikäinen et
al., 2005), higher organic C/N (Chen et al., 2015) and a more arid soil environment
(Marusenko et al., 2013) compared to bacteria. Climate warming and grazing can
change vegetation cover, soil water and energy balance, alter the quantity and quality
of soil organic matter and mineral N content (Saggar et al., 2004), and thus affect $N_2O$
production (Shi et al., 2017). However, it remains unknown how bacteria and fungi
respond to concurrent warming and grazing and contribute to $N_2O$ production in alpine
grasslands.
To clarify whether fungi played the mainly role in $N_2O$ production process and its
response to warming and winter grazing in alpine grasslands, we used a warming and
grazing experiment over 10 years in an alpine meadow on the Tibetan Plateau. We
measured the gene abundance of soil bacterial and fungal communities using
quantitative PCR, and the potential of $N_2O$ emission from bacterial and fungal
nitrification and denitrification through an incubation experiment to assess the
contribution of $N_2O$ production potential from bacteria and fungi. We aimed to test the
following hypotheses: (1) soil fungi were the main contributor to $N_2O$ production
because of the low soil temperature and high organic C and N in the alpine grasslands,
and (2) although $N_2O$ emission was not affected by warming and winter grazing at our
site (Zhu et al., 2015), the biotic pathways responsible for $N_2O$ would be changed due
to the distinct preferred soil environments of bacterial and fungal communities.

**2 Materials and Methods**
**2.1 Site and sampling.** Details of the experimental site and design of the warming and
grazing were described by Wang et al. (2012). The experiment was conducted in an
alpine grassland (37°37'N, 101°12'E, 3250 m elevation) at the Haibei Alpine Meadow
Ecosystem Research Station of the Chinese Academy of Sciences. Over the past 25
years, the mean annual temperature was -2 $^{\circ}$C, and the mean annual precipitation was
500 mm. In soil sampling year and month of 2015, mean temperature was 0 $^{\circ}$C and 9.7
$^{\circ}$C, respectively; total rainfall was 327.2 mm and 46.6 mm, respectively. Over 80% of
total rainfall falls during the summer monsoon season (Luo et al., 2010; Zhao and Zhou,
1999). The soil was classified as Gelic Cambisols (WRB, 1998). The plant community
at the experimental site is dominated by *Kobresia humilis*, *Festuca ovina*, *Elymus*
*nutans*, *Poa pratensis*, *Carex scabrirostris*, *Gentiana straminea*, *Gentiana farreri*,
*Blysmus sinocompressus*, *Potentilla nivea* and *Dasiphora fruticosa* (Luo et al., 2010).

A two-way factorial design (warming and grazing) was used with four replicates of
each of four treatments (Wang et al., 2012), beginning in May 2006, namely no
warming with no grazing (C), no warming with winter grazing (G), warming with no
winter grazing (W) and warming with winter grazing (WG). In total, 16 plots of 3-m
diameter were fully randomized throughout the study site.
For warming treatments, the design of the controlled warming (i.e. free-air
temperature enhancement (FATE) system with infrared heaters) with grazing
experiment described previously by Kimball et al. (2008) and Wang et al. (2012). Free-
air temperature enhancement using infrared heating has been set up to create a warming
treatment since May 2006 (Luo et al., 2010). The differences in canopy temperature at
set points between heated plots and the corresponding reference plots were 1.2°C during
the daytime and 1.7°C at night in summer. During winter, from October to April, the
power output of the heaters was manually set at 1500 W per plot to make sure the
increased of soil temperature was the same with it in summer, as some infrared
thermometers were not working.
For grazing treatments, summer grazing treatments were used to explore the effects
of warming and grazing on ecosystem during the growing season from 2006 to 2010
(Luo et al. 2010; Hu et al. 2010; Wang et al. 2012). Considering strong disturbance,
grazing was stopped during 2011-2015, summer grazing was replaced by cutting and
removing about 50% of the litter biomass in October and the following March each
year to simulate winter grazing. Given the importance of winter grazing, winter grazing
during the non-growing seasons was further investigated (Zhu et al. 2015; Che et al.
2018). Alpine meadows in the region can be divided into two grazing seasons (i.e.,
warm-season grazing from June to September and cold-season grazing from October to
May) (Cui et al., 2015). Before the experiment was conducted, we had examined how
clipping simulated the effects of actual grazing before we established four replicated
"actual grazing treatments" compared with the "simulated grazing treatments", the soil
and plant all showed no difference between simulated grazing and actual grazing
treatments (Klein et al. 2004; 2007), because the soil is frozen in winter, meaning that
the effect of selective feeding and trampling by sheep would be limited, so the effect of
cutting in winter was similar to winter grazing (Zhu et al., 2015).
**2.2 Soil sampling.** Five soil cores (5 cm in diameter) were randomly collected
within each plot on 15 August 2015 at a depth of 0–20cm (including organic layer) and
then mixed to form a composite sample. All soil samples were transported to the
laboratory and sieved through a 2-mm mesh before being stored at -20ºC or 4ºC for
further molecular analyses.

**2.3 Soil properties and gene abundance of bacteria and fungi analysis.** Soil
moisture content was measured by drying at 105°C for 24 hours. For soil mineral N
($NH_4^+$-N and $NO_3^-$-N) analyses, 10 g of soil (field-moist) was shaken for 1 hour with
50 mL of 1 M KCl and filtered through filter paper, and determine the $NH_4^+$-N and $NO_3^-$
-N concentrations by Skalar flow analyzer (Skalar Analytical, Breda, The Netherlands).
Total C and N content were measured by using combustion elemental analyzers
(PerkinElmer, EA2400, USA).
Soil DNA was extracted from 0.5 g of frozen soil using a FastDNA™ Kit for Soil
(QBIOgene) based on the instructions and stored at -20°C. Total bacteria and fungi
copies were quantified by real-time PCR using an iCycler thermal cycler equipped with
an optical module (Bio-Rad, USA)
The real-time PCR mixture contained 5 ng of soil DNA, 2 pmol of primers and 10×iQ
SYBR Green super mix (Bio-Rad), in a 20-μL reaction volume. The primer for bacteria
were     341F     5'-CCTACGGGAGGCAGCAG-3'     and     534R     5'-
ATTACCGCGGCTGCTGGCA-3' (Muyzer et al., 1993). The thermal cycle conditions
were 10 min at 95°C; 35 cycles of PCR were then performed in the iCycler iQ Real-
Time PCR Detection System (BIORAD) as follows: 20 s at 95°C, 15 s at 55°C and 30
s at 72°C. A final 5-min extension step completed the protocol. The primer for fungi
were FU18S1 5'-GGAAACTCACCAGGTCCAGA-3' derived from Nu-SSU-1196
and Nu-SSU-1536 5'-ATTGCAATGCYCTATCCCCA-3' (Borneman and Hartin, 2000)
and the thermal cycle conditions were one step of 10 min at 95°C, then 40 cycles of
PCR performed as follows: 20 s at 95 °C, 30 s at 62 °C and 30 s at 72 °C. A final 5-min
extension step completed the protocol.
**2.4 Total, fungal and bacterial nitrification enzyme activity, and total, fungal and**
**bacterial potential of $N_2O$ production from denitrification analysis.**
Fungal (FNEA), bacterial (BNEA) and total nitrification enzyme activity (TNEA)
were determined following the protocol described in Dassonville et al. (2011). Briefly,
moist field soil equivalent to 12 g of dry soil was weighed into 240-mL specimen bottles
(LabServ), 12 mL of $NH_4$-N solution (50 µg N-$(NH_4)_2SO_4$ $g^{-1}$ dry soil) and distilled
water was added to achieve a 96 mL total liquid volume, and the slurry was incubated
at 28°C for 10 hours with constant agitation (180 rpm) in an orbital shaker (Lab-Line
3527; Boston, MA, USA) to mix slurry well and provide an aerobic environment. Three
treatments were imposed: (I) cycloheximide ($C_{15}H_{23}NO_4$, a fungicide) at 1.5 mg $g^{-1}$ in
solution was used to inhibit the nitrification activity from soil fungi, (II) streptomycin
sulphate ($C_{42}H_{84}N_{14}O_{36}S_3$, a bactericide) at 3.0 mg $g^{-1}$ in solution was used to inhibit
the nitrification activity from soil bacteria (Castaldi and Smith, 1998;Laughlin et al.,
2009) and (III) a no-inhibitor control was used to show the total nitrification activity.
During incubation, 10 mL of the soil slurry was sampled with a syringe at 2, 4, 6, 8 and
10 h, and then filtered through Whatman No. 42 ashless filter paper. Filtered samples
were stored at -20 °C until analysis for $NO_2^-+NO_3^-$ concentration on a LACHAT
Quickchem Automated Ion Analyzer (Foss 5027 Sampler, TECATOR, Hillerød,
Denmark). Linear regression between the $NO_2^-+NO_3^-$ production rate and time was
observed, and the rates of nitrification enzyme activity were determined from the slope
of this linear regression. The nitrification enzyme activity of soil fungi was estimated
by the difference between rates of nitrification enzyme activity under treatment (III)
and treatment (I); the nitrification enzyme activity of soil bacteria was estimated by the
difference between rates of nitrification enzyme activity under treatment (III) and
treatment (II). The total nitrification enzyme activity was from treatment III.
Fungal (FDEA), bacterial (BDEA) and total potential of $N_2O$ production (TDEA)
from denitrification was measured in fresh soil from each plot following the protocol
described in Patra et al. (2006) and Marusenko et al. (2013). Three sub-samples
(equivalent to 12 g dry soil) from each soil sample were placed into 240-mL plasma
flasks, and 7 mL of a solution containing $KNO_3$ (50 μg $NO_3^-$N $g^{-1}$ dry soil), glucose
(0.5 mg C $g^{-1}$ dry soil) and glutamic acid (0.5 mg C $g^{-1}$ dry soil) were added. Additional
distilled water was provided to achieve 100% water-holding capacity and optimal
conditions for denitrification. Three treatments were imposed: (I) cycloheximide
($C_{15}H_{23}NO_4$; a fungicide) at 1.5 mg $g^{-1}$ in solution was used to inhibit the fungal
potential of $N_2O$ production from denitrification, (II) streptomycin sulphate
($C_{42}H_{84}N_{14}O_{36}S_3$; a bactericide) at 3.0 mg $g^{-1}$ in solution was used to inhibit the bacterial
potential of $N_2O$ production from denitrification (Castaldi and Smith, 1998; Laughlin
and Stevens, 2002), and (III) a no-inhibitor control was used to show the total potential
of $N_2O$ production from denitrification. The headspace air of the specimen bottles was
replaced with $N_2$ to provide anaerobic conditions. Specimen bottles were then sealed
with a lid containing a rubber septum for gas sample collection. Specimen bottles with
the soil slurry were then incubated at 28°C for 48 h with constant agitation (180 rpm)
in an orbital shaker (Lab-Line 3527; Boston, MA, USA). During incubation, 12-mL gas
samples was taken at 0, 24 and 48 h with syringes and injected into pre-evacuated 6-
mL glass vials. The $N_2O$ concentration of the gas samples was analyzed via gas
chromatography. The potential of $N_2O$ production from denitrification were calculated
from the slope of the regression using values for 0, 24 and 48 hours of incubation. The
fungal potential of $N_2O$ production from denitrification was estimated by the difference
between potential production under treatment (III) and treatment (I); The bacterial
potential of $N_2O$ production from denitrification was estimated by the difference
between rates of denitrification enzyme activity under treatment (III) and treatment (II).
Total denitrification enzyme activity was from Treatment III.
For the contribution of bacteria and fungi to total nitrification enzyme activity was
calculated it by the ratio of BNEA or FNEA to BNEA+FNEA; the contribution of
bacteria and fungi to total potential of $N_2O$ production from denitrification was
calculated it by the ratio of BDEA or FDEA to BDEA+FDEA.
**2.5 Statistical analysis.**    For the controlled experiment, the statistical significance of
the effects of warming, grazing and their interaction on plant biomass, soil properties,
microbial functional genes, and fungal and bacterial nitrification enzyme activity and
potential of $N_2O$ production from denitrification were tested by two-way ANOVA in
the PROC GLM procedure of SAS (version 9, SAS Institute, Cary, NC, USA).

**3 Results**
**3.1 Plant biomass and soil properties**
The average plant standing biomass was 343, 345, 301 and 362 g dry matter $m^{-2}$ in
the control, G, W and WG treatments measured at the day of soil sampling, respectively.
Grazing and warming had no effect on plant biomass (Fig. 1a, Table 1).
Soil temperature varied from 11.8 to 14.0 °C. Grazing (P=0.05) and warming (P<0.01)
increased soil temperature (Fig. 1b, Table 1). The average soil moisture varied from 26%
to 34% (w/w). Grazing had no effect on soil moisture, which was lower in warming
plots (P<0.01) (Fig. 1c, Table 1). There was an interactive effect between grazing and
warming on soil temperature (P<0.01).
Soil total C (TC) was not affected by grazing (P=0.13) or warming (P=0.12) alone,
but there was a marginal interaction between grazing and warming on TC (P=0.07) (Fig.
2a, Table 1). Similar to TC, soil total N (TN) also showed no response to grazing or
warming (Fig. 2b, Table 1). Soil $NH_4^+$-N content was lower in warming treatments than
in no-warming treatments (P=0.05) (Fig. 2c, Table 1). Greater soil $NO_3$-N content
occurred under the warming treatments (P=0.05) than under the no-warming treatments
(Fig. 2d, Table 1).

## 3.2 Microbial functional genes

Bacterial gene abundance varied from $4.71 \times 10^9$ to $5.93 \times 10^9$ copies $g^{-1}$ dry soil, which was much higher than fungal gene abundance (Fig. 3). Warming and grazing both increased the bacterial gene abundance in soil ($P<0.01$), but there was no interaction effect between them on bacterial gene abundance (Table 1). By comparison, fungal gene abundance showed no difference across all treatments.

## 3.3 Nitrification enzyme activity and potential of N$_2$O production from denitrification of bacteria and fungi

TNEA varied from 1.07 to 1.64 μg N $g^{-1}$ $h^{-1}$ in all treatments. BNEA ranged from 0.43 to 0.64 μg N $g^{-1}$ $h^{-1}$, which was lower than the FNEA in soil (0.59–0.66 μg N $g^{-1}$ $h^{-1}$) ($P=0.01$) (Fig.4 a-c). FNEA was lower under warming treatments than under the no-warming treatments ($P=0.05$) (Table 1).

TDEA was between 1.32 and 1.80 μg N $g^{-1}$ $h^{-1}$. FDEA was clearly the dominant process for TDEA (Fig. 4 d-f), because it was higher than BDEA for all treatments except warming. Warming increased BDEA ($P=0.04$). Warming and grazing had a significant interaction effect on FDEA ($P<0.01$) (Table 1).

## 3.4 The contribution of bacteria and fungi to potential N$_2$O emissions

The contribution of FNEA to TNEA varied from 47 ± 2% to 56 ± 5%, and the contribution of FDEA to TDEA varied from 45 ± 3% to 63 ± 3% (Fig. 5). Warming

significantly decreased the contribution of FNEA and FDEA to TNEA and TDEA in
soils (FNEA: P=0.02; FDEA: P=0.04).

**4 Discussion**
$N_2O$ is mainly produced from microbial nitrification and denitrification processes,
but the contribution of bacteria and fungi to nitrification and denitrification processes
is still unclear. In our results, fungi contributed 54% and 63% of the TNEA and TDEA,
respectively, in control treatment of the alpine grassland studied. Our result of the
fungal contribution to potential of $N_2O$ production is much lower than Laughlin and
Stevens (2002) and Zhong *et al.* (2018) whom reported 89% and 86% fungal
contribution from temperate grasslands, but is higher than the 40-51% fungal
contribution observed across different ecosystems by Chen et al. (2014). Kato et al.
(2013) also showed that $N_2O$ emissions from FDEA were higher than from BDEA in
alpine meadows, reinforcing the important role fungi play in the denitrification process.
Our findings support our first hypothesis and further proved that both nitrification and
denitrification were largely driven by fungal communities in alpine meadow grasslands.
A possible explanation is that fungi prefer the arid, high organic substrate and low-
temperature environment (Pietikäinen et al., 2005; Chen et al., 2015; Marusenko et al.,
2013). In alpine grasslands, the mean annual temperature is 0 $^{\circ}$C; even during the
sampling day the mean temperature was only 11 $^{\circ}$C. The cold environment could cause
higher activity in fungi than in bacteria. Moreover, the cold environment decreases the
rate of mineralization, leading to greater organic C and N accumulation (Ineson et al.,
1998;Schmidt et al., 2004). In our study, soil TC and TN concentrations were 72–86 g
kg$^{-1}$ and 6–7 g kg$^{-1}$, respectively (Fig. 2a and 2b), much higher than in temperate
grasslands and farmland, providing a favorable environment for fungi (Bai et al., 2010).
These are mainly reasons that soil fungi played the mainly role in $N_2O$ production
process in the Tibetan alpine grasslands.
Our methodology did not exclude a role for archaea in nitrification and denitrification.
Previous studies on grasslands only focused on fungal and bacterial process because
archaeal specific inhibitors have not yet been identified for N cycling processes.
However, archaea are widespread in soil, are involved in nitrification denitrification
(Cabello et al., 2004), eg. archaeal ammonia oxidizers are globally (Leininger et al.,
2006). In our study, we also found the TNEA was higher than the sum of NEA from
bacteria and fungi, while TDEA was higher than DEA from bacteria and fungi (Fig. 4),
which showed that archaea also played the role in $N_2O$ production process in our site.
The development of inhibitor-based approaches may help to show how archaea
responses to environmental change (Marusenko et al. 2013).
Our results supported the second hypothesis that although warming did not change
the total $N_2O$ production potential on the Qinghai-Tibetan Plateau, the biotic pathways
responsible for $N_2O$ had been changed, as bacterial contribution to TNEA and TDEA
all were higher than fungal which suggested the higher bacterial $N_2O$ production
potential under warming treatment (Fig. 4, Table 1). The increase in bacterial $N_2O$
production potential, coupled with a decrease in fungal $N_2O$ production potential, could
be the main reasons why the total $N_2O$ production potential was no difference between
control and warming treatments. The field data of $N_2O$ emission in our site was
measured in the year of 2011−2012 also showed no effect of warming on $N_2O$ emission
(Zhu et al. 2015). Our results reinforced this and suggested that bacterial nitrification
and denitrification process alone is unable to accurately describe the response of $N_2O$
to warming.

352       It is the two reasons that lead to the changes of fungal and bacterial pathways for

$N_2O$ production process by warming. Firstly, warming significantly increased the soil
temperature (Fig.1b, Table 1), the increased of soil temperature directly reduces fungal
activity but increase bacterial activity, because fungi prefer the cold environment
compared with bacteria (Pietikäinen et al., 2005). Secondly, fungi prefer higher organic
C/N environment while bacteria prefer higher inorganic C/N environment (Chen et al.,
2015). In our site, although the soil $NH_4^+$-N concentration did not change with warming,
soil $NO_3^-$-N concentration was significantly increased showed the soil inorganic N was
increased (Fig. 2a and 2b, Table 1); on the other hand, the soil dissolved organic
nitrogen was significantly decreased from 48 to 41 mg $kg^{-1}$ ($P<0.04$), the soil labile C
and N was also found significantly decreased by warming (Rui et al., 2012), it showed
the soil organic C and N was decreased in our site. Therefore, warming indirectly reduce
fungal activity but increase bacterial activity through increased soil inorganic N and
decreased soil organic N in our site. In our site, the FNEA and FDEA were reduced by
16% and 30% respectively, but the BNEA and BDEA were increased by 15% and 41%
respectively by warming. All these changes resulted in fungi contributing less to
nitrification and denitrification than bacteria (Fig.5). Although the gene abundance of
fungi was not changed by warming which showed inconsistencies with the changes of
FNEA and FDEA, these inconsistencies might be explained by the fungal gene
abundance was not likely provided information on real-time process rates since such
rates are dependent on environmental conditions, fluctuations in environmental
conditions can cause rapid changes in real-time process rates, but not necessarily affect
gene abundance (Zhong et al. 2014). In summary, it indicates that the soil microbial
process was altered by warming, even though the total potential of $N_2O$ production did
not change, with a shift in dominance from fungi to bacteria in $N_2O$ production process
after 10 years of warming.
Numerous studies have demonstrated that grazing can impact microbial processes
and induce the loss of N through: (1) altering the substrate concentration for $N_2O$
production and reduction in soil through the deposition of dung and urine (Saggar et al.,
2004); (2) reducing vegetation cover due to changes in soil water content and energy
balance (Leriche et al., 2001); and (3) increasing soil compaction and reducing soil
aeration through animal tramping (Houlbrooke et al., 2008). However, most of these
were focused on grazing in the growing season, little was focused on the effect of winter
grazing on N cycle process. In this study fungal and bacterial potential of $N_2O$
production from nitrification and denitrification all showed little response to winter
grazing (Fig. 4, Table 1). A possible explanation is that neither soil moisture, plant
biomass nor organic/inorganic C/N content was affected by winter grazing (Fig.1-2,
Table 1). Additionally, the soil was frozen in winter, so that the effect of selective
feeding and trampling could be limited by grazing sheep (Zhu et al., 2015;
Krümmelbein et al., 2009; Steffens et al., 2008). As a result, the same soil
environmental conditions for both winter grazing and control had no effect on soil fungi
and bacteria, and thus on fungal and bacterial nitrification and denitrification. Moreover,
the field data of $N_2O$ emission in the year of 2011-2012 also support the results and
suggest that replacing summer grazing by winter grazing could cause the soil N cycle
process to become stable (Zhu et al. 2015).

397        Overall, we conclude that fungi played the dominant role in the $N_2O$ production

process in alpine meadows. Previous study had proved the climate warming did not
affect the $N_2O$ production in our site (Zhu et al. 2015), but we found warming could
alter biotic pathways responsible for $N_2O$ production process on the Tibetan Plateau.
Our study exhibited the effects of a decade of the simulation experiment; however, a
thorough understanding about the long-term impact of warming and grazing on soil
fungal nitrification and denitrification from alpine meadow grassland requires further
investigation for a multi-decade period.

405        From this study, due to the different adaptation strategies of fungi and bacteria, and

their different nutrition requirements, future changes in climate and soil resources are
likely to affect biogeochemistry in a way not currently accounted for in ecosystem
models that assume N transformations are controlled only by bacteria. Accurate
predictions for $N_2O$ production and N loss due to environmental change and land use
will benefit from the inclusion of fungi as key mediators of ecological processes in
grasslands.


## Competing interests

The authors declare that they have no conflict of interest.

## Acknowledgements

This work was supported by the National Key R&D Program of China (No.

2016YFC0501802), the National Natural Science Foundation of China (No. 41601245;

31672474), the Foundation of Committee on Science and Technology of Tianjin

(No. 16YFXTSF00500), and supported by the Strategic Priority Research Program B

of the Chinese Academy of Sciences (No. XDB15010201). We also thank Miss Ri Weal

for her assistance in improving the use of English in the manuscript.

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

Table 1. Results (F-value and P-value) from two-way ANOVA for the effects of
warming (W), winter grazing (G) and their interactions (WG) on soil and microbial
characteristics.



|  | W | | G | | WG | |
|---|---|---|---|---|---|---|
|  | F value | P value | F value | P value | F value | P value |
| Biomass | 0.21 | 0.65 | 1.41 | 0.26 | 1.21 | 0.29 |
| Temperature | 61.16 | **<0.01** | 4.64 | **0.05** | 25.54 | **<0.01** |
| Soil moisture | 14.87 | **<0.01** | 0.17 | 0.68 | 0.13 | 0.72 |
| TC | 2.69 | 0.12 | 2.7 | 0.13 | 3.95 | 0.07 |
| TN | 1.44 | 0.25 | 1.47 | 0.25 | 3.02 | 0.11 |
| $NH_4^+$-N | 4.57 | **0.05** | 1.6 | 0.23 | 0.02 | 0.89 |
| $NO_3^-$-N | 3.6 | 0.05 | 1.42 | 0.25 | 0.09 | 0.81 |
| Bacteria | 17.91 | **<0.01** | 11.67 | **<0.01** | 0.11 | 0.75 |
| Fungi | 1.72 | 0.21 | 0.70 | 0.42 | 2.89 | 0.12 |
| BNEA | 1.01 | 0.90 | 3.24 | 0.35 | 3.94 | 0.07 |
| FNEA | 4.58 | **0.05** | 1.15 | 0.34 | 0.37 | 0.51 |
| TNEA | 0.8 | 0.39 | 2.23 | 0.16 | 0 | 0.95 |
| BDEA | 5.16 | **0.04** | 2.45 | 0.14 | 4.04 | 0.07 |
| FDEA | 1.52 | 0.24 | 0.96 | 0.34 | 9.98 | **<0.01** |
| TDEA | 0.98 | 0.34 | 2.33 | 0.15 | 0.15 | 0.70 |

Bold indicates significance at $P < 0.05$.


**Figure caption**



**Fig. 1**, Plant biomass (a) soil temperature (b) and soil moisture content (c) in an alpine
meadow. C (■), control treatment; G (□), winter grazing treatment; W (▦), warming
treatment; WG (▦), warming combined with the winter grazing treatment. Values are
means ±1 s.e.m. (*n*=4).


**Fig. 2** Soil total carbon (TC) (a), soil total nitrogen (TN) (b), soil $NH_4^+$-N (c) and $NO_3^-$
-N (d) content in an alpine meadow. C (■), control treatment; G (□), winter grazing
treatment; W (▦), warming treatment; WG (▦), warming combined with the winter
grazing treatment. Values are means ±1 s.e.m. (*n*=4).

**Fig. 3** Abundance of bacteria (a) and fungi (b) in an alpine meadow; C (■), control
treatment; G (□), winter grazing treatment; W (▦), warming treatment; WG (▦),
warming combined with the winter grazing treatment. Values are means ±1 s.e.m. (*n*=4).

**Fig. 4** Bacterial nitrification enzyme activity (BNEA) (a), fungal nitrification enzyme
activity (FNEA) (b), total nitrification enzyme activity (TNEA) (c); Bacterial potential
of $N_2O$ production from denitrification (BDEA) (d), fungal potential of $N_2O$ production
from denitrification (FDEA) (e) and total potential of $N_2O$ production from
denitrification (TDEA) (f) in an alpine meadow. C (■), control treatment; G (□),
winter grazing treatment; W (▦), warming treatment; WG (▦), warming combined
with the winter grazing treatment. Values are means ±1 s.e.m. (*n*=4).

**Fig. 5** Contribution of bacteria and fungi to total nitrification enzyme activity (box with
the red and dashed line) and total potential of $N_2O$ production from denitrification (box
with the black and solid line) in an alpine meadow. C (■), control treatment; G (□),
winter grazing treatment; W (▦), warming treatment; WG (▦), warming combined
with the winter grazing treatment. Values are means ±1 s.e.m. (*n*=4).




Fig.1

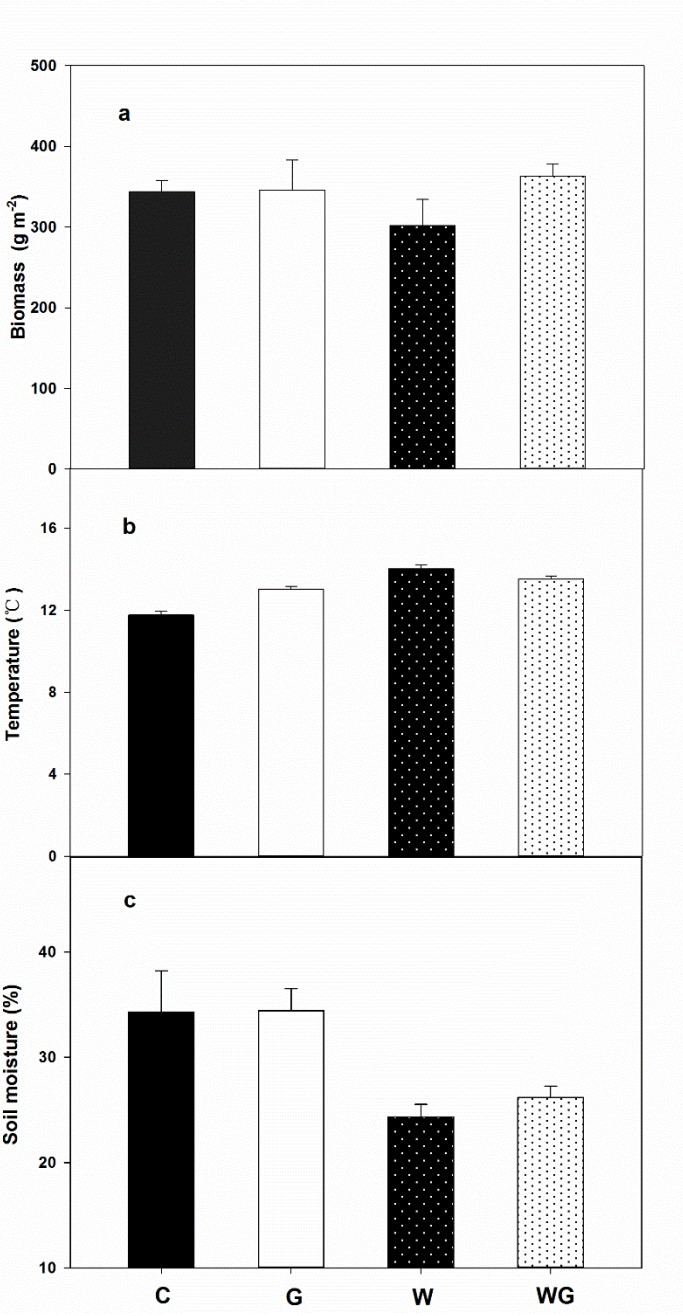


Fig. 2

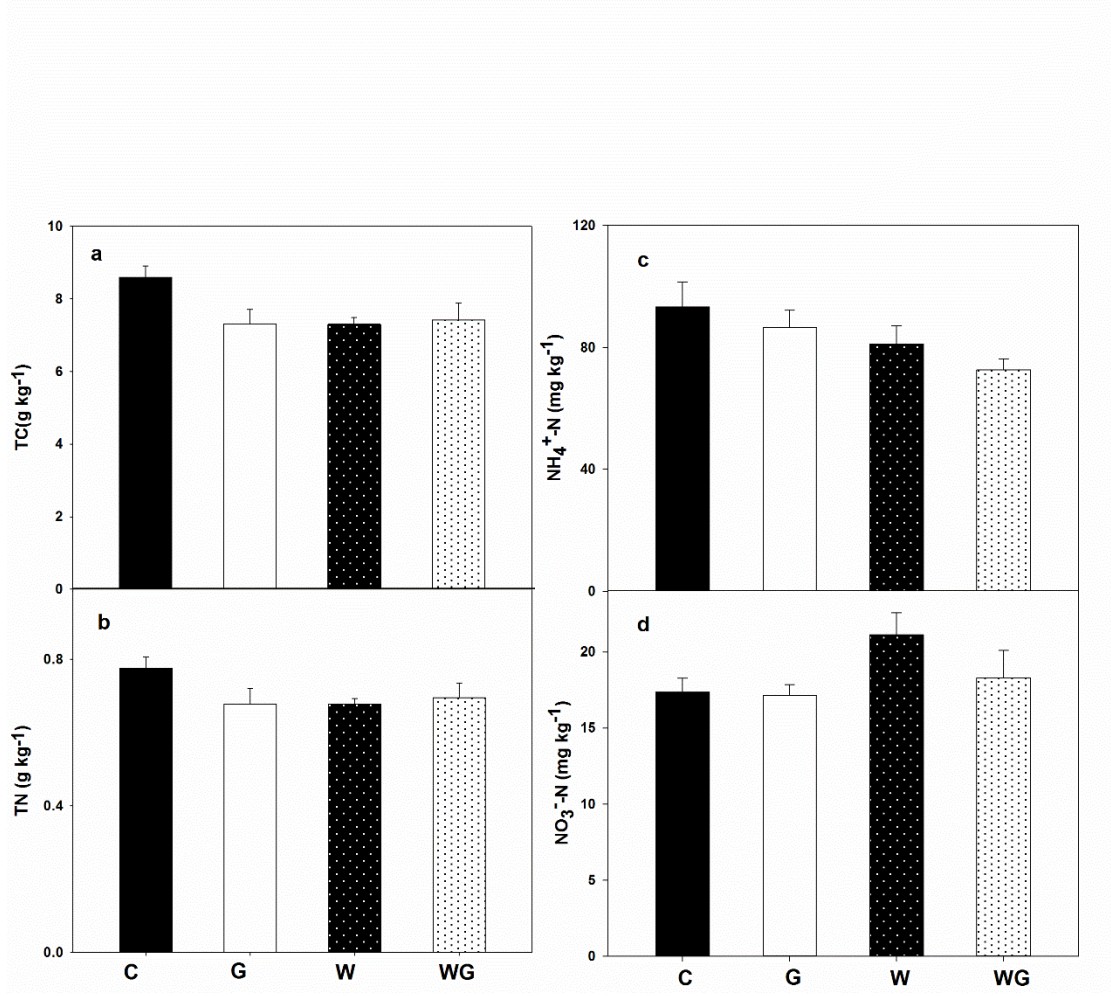


Fig. 3

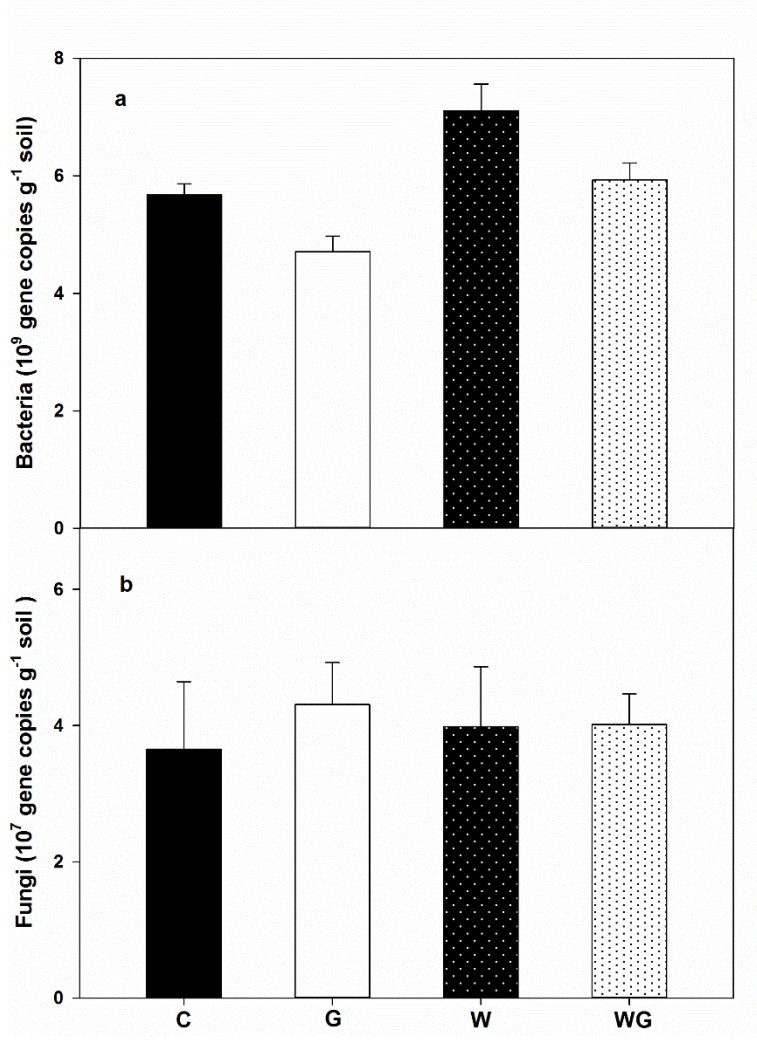


Fig.4

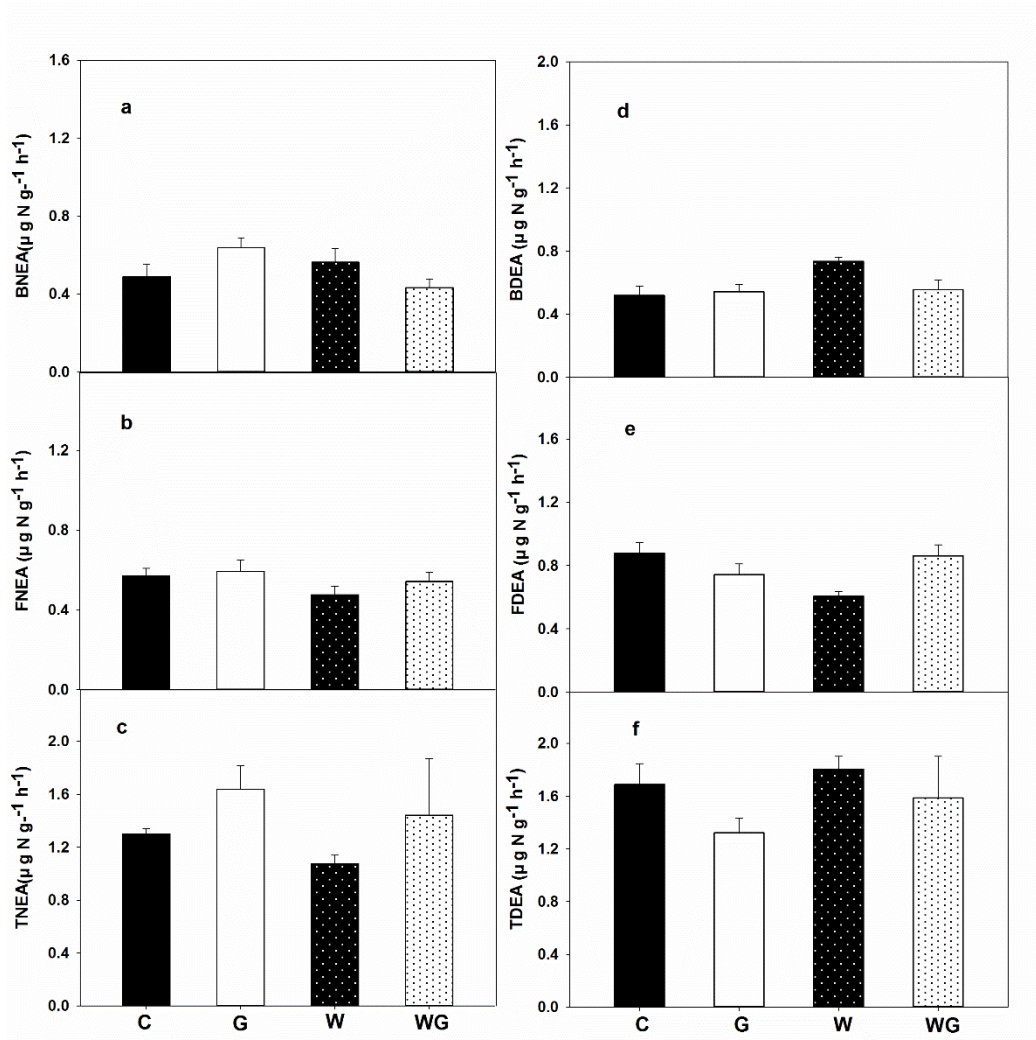


Fig.5

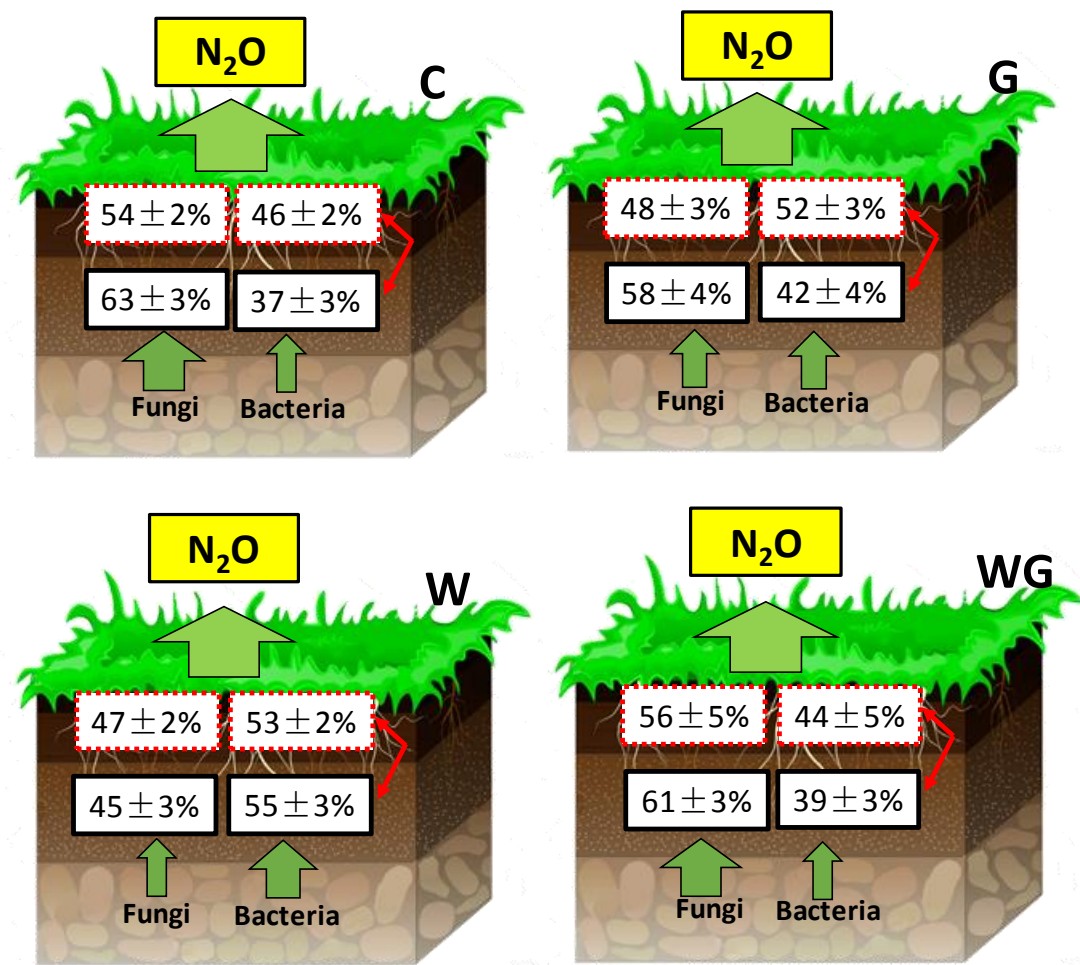
