# Peer review of "Fungi regulate response of N2O production process to warming and"

_Biogeosciences, 2017_

## Referee Comment (RC1) · X. Feng (Referee) · 19 Mar 2018

This is a concise and nicely written paper, focusing on fungal and bacterial contributions to potential N2O emissions in an alpine grassland in response to warming and grazing treatments in the field. The authors report several interesting observations, including an increased bacterial enzyme activity and a decreased fungal enzyme activity for N2O emissions under warming. The results have immediate implications for GHG emissions under the scenario of climate change. I have several suggestions for the authors to consider in order to improve the manuscript.

1. Although the authors showed that fungal and bacterial pathways for N2O emissions changed in different directions under warming, the underlying mechanisms or causes

remain unknown. In Line 321-322, it is mentioned that increased NO3-N may inhibit fungal growth. Can you elaborate more? Also, did warming affect soil moisture contents or dynamics compared to the control? If so, how would moisture change affect fungal versus bacterial communities? In the end, I am interested in the driving force leading to the observed changes—it is direct warming effect or indirect effect mediated by other factors? Unless we know answers to these questions, we can hardly speculate on the future changes.

2. Speaking of future predictions, I think it should be emphasized that measurements made here were potential rather than "real" emissions in the field. A critical requirement for denitrification to occur is anoxic or sub-oxic conditions. Therefore, I would think that N2O emissions more depend on the hydrological or redox conditions of the soil. Observations of fungal and bacterial enzyme activity changes in the lab may or may not apply to the field observations, depending on how warming affects soil moisture.

Some minor points: Line 163: I notice that there was no field replicate for the measurement?

Line 223: N2 not N.

Line 227: Why only three time points for the denitrification measurement versus 5 points for nitrification?

Lines 285 and 292: NEA, DEA, FDEA, BDEA. . .not used in the previous text.

Line 304: I don't think IC is much higher in Haibei soils than some temperate grassland soils in Mongolia. IC contents are dependent on soil pHs. . .

---

## Referee Comment (RC2) · Anonymous Referee #3 · 29 Mar 2018

This study reports the effect of warming and grazing on soil biotic contribution to N2O production in a Tibetan grassland, by examining a long-term (over 10 years) experiment combined with an incubation experiment. Their results indicated that fungi could be the main source for N2O production potential in the Tibetan alpine grasslands. Overall, the manuscript is of interest and generally well written. But there are some concerns and unclear points that should be addressed prior to publication.

Please find some more detailed comments below.

1. Lines 162-164, is it enough to collect only 5 cores for each soils?

2. How the authors draw the contribution of bacteria and fungi to total nitrification enzyme activity and total denitrification enzyme activity as shown in Fig. 5? I cannot

find the specific description in the section "Materials and Methods".

3. Line 257, "Fig. 1A" should be changed to "Fig. 1a", based on the Figure 1. Also, the authors should revised it throughout the main text.

4. Line 259, "soil moisture" should be changed to "The average soil moisture".

5. In Figs. 1 and 4, why significant differences were only shown in Figs. 1b and 4e rather than all of subfigures?

---

## Referee Comment (RC3) · Anonymous Referee #1 · 7 Apr 2018

Comments on Zhong et al. for Biogeosciences Discussion

This manuscript presents an interesting study on the response of an alpine grassland ecosystem to warming and grazing in the period of 10 years. N2O production via variable microbial components was the main focus. It is written concisely and easy to understand. However, regarding the experiment design and interpretation of the dataset, I believe that there is still more to improve before it could be published. Despite their investigation into multiple treatments and parameters, the authors need to provide more field evidence and literature comparison to reach a convincing conclusion.

Throughout the whole manuscript, the authors seem to mix up denitrification enzymatic activity and N2O production. If the inhibitors applied in the experiments to determine denitrification rates also inhibit N2O reduction to N2, the N2O production should rather represent potential denitrification rates. If N2O reduction was not inhibited during the experiment, the results could not be noted as "denitrification rates". Please clarify this key point and make revision accordingly. The methods determining these rates should be described in more details in M&M.

Line 111: "To clarify whether fungi control the N2O production process" is misleading as Fungi contributes anyway; I assume that the authors wish to clarify the "role of fungi in N2O production process"

Line 161-162: Please explain this; why do you see the effects on ecosystem level despite that plot size are 3 m? Any data to support this?

Line 165: If 10 years' warming and grazing treatment was done, why was only one sampling of soils by the end of 10 years' treatment? Have you considered the soil heterogeneity between control and treatment plots since the beginning of treatments?

Line 166: Including or excluding organic layer? Please specify.

Line 225-226: 100% of water-holding capacity could favor denitrification; however, it may not likely represent field condition, which is usually drier. Please justify your choice of such incubation condition.

Line 294-298: Use present tense: use "is" to replace "was".

Line 298: Change "who" to "whom".

Lne 314 to 315: When comparing the studied alpine grassland to temperate grassland, how do come to to the conclusion that the lower inorganic C and N contents in soil were due to larger fungal contribution to N2O production? What about the higher mineralization rates in the temperate systems? In addition, the control of inorganic C or N levels in soil could be also related to biomass uptake and turnover. Please clarify it and avoid such speculation.

Line 324: "common" and "globally" do not fit together; please revise.

Line 348-349: "gene abundance of fungi was not changed" against treatments; how do you reconcile your finding with the hypothesis?

---

## Referee Comment (RC4) · Anonymous Referee #5 · 8 Apr 2018

I have some major concerns as shown below: 1. The experimental design is not acceptable. Firstly, why did you choose "winter grazing"? There seems no explanation. The temperature should be too low to let the animal grazing out of the field in winter. Additionally, the grassland is expected to be covered by snow and the grasses should be withered in winter. Secondly, the description of the treatment is really confusing. Winter grazing should be used in the current study, but "For grazing treatments, the grazing treatments in this site were used for summer grazing treatments until 2010, from 2011 to 2015, there was no grazing during the summer, and grazing was replaced by cutting and removing about 50% of the litter biomass in October and the following March each year to simulate winter grazing" (lines 153-156). To be honest, I can't understand the experimental design at all. In addition, grazing can't be simulated by

cutting or mowing, since grazing involves tread and urine/dung deposition. Even the land is very hard due to freezing in winter, tread by animals would result in different effects on the plant communities. 2. I can't see how you can jump from nitrification or denitrification potentials to assessing the contributions of bacterial and fungi to potential N2O emissions. Nitrification or denitrification potentials should not be regarded as N2O productions especially emissions by nitrification or denitrification. From this sense, the discussion section should be rewritten thoroughly. 3. The manuscript is not well prepared. There are lots of writing issues throughout the manuscript. I only presented few of them since there are too many.

Abstract Lines 44-46: The treatments should be described briefly in the abstract to increase the readability. Additionally, some key information about the method should be presented. Lines 46-52: The values should be presented with uncertainties, e.g., standard error, standard deviation or 95% confidence interval. Similarly, the relevant values in the text should be presented with uncertainties. Lines 46-47: Were these values got from the control? Lines 49-52: Suggest rephrase these sentences in such way: "However, warming significantly increased the enzyme activity of bacterial nitrification and denitrification to 53% and 55%, respectively, but decreased enzyme activity of fungal nitrification and denitrification to 47% and 45%, respectively. Winter grazing had no such effects." Lines 52-54: How could you make this conclusion? Under what conditions do soil fungi contribute more to N2O production? This sentence is of course not clear. If the conclusion is obtained based on results from the control, it should be put somewhere after lines 46-47. Additionally, can you make such a strong conclusion based on an incubation experiment? Lines 56-58: This should not be put in the abstract as a key implication since it should be regarded as a fact. Line 59-60: This sentence should be rephrased since some grammar issue exists. For example, "lead to refine…." is not correct. Overall, the abstract needs substantial revision.

Introduction Line 66: not clear what does "it" refer to. Lines 67-69: This sentence needs substantial revision. Line 122: Why did you choose "winter grazing"? There seems no

explanation. The temperature should be too low to let the animal grazing out of the field in winter. Additionally, the grassland is expected to be covered by snow and the grasses should be withered in winter.

M & M Lines 130-131: The symbol °C is not correctly used. Lines 131-132: over 80% of which? Lines 133-134: Please clearly present the soil classification systems and the references. Lines 134: There should be a space between the word and the parentheses here and in other sentences or Figures (Please check the figures as well). Line 139: The indent here is not consistent with other paragraphs. Please keep consistency. Line 146: delete was. Lines 153-156: The description is really confusing. According to the above paragraph, winter grazing was used in the current study, but "For grazing treatments, the grazing treatments in this site were used for summer grazing treatments until 2010, from 2011 to 2015, there was no grazing during the summer, and grazing was replaced by cutting and removing about 50% of the litter biomass in October and the following March each year to simulate winter grazing". To be honest, I can't understand the experimental design at all. In addition, grazing can't be simulated by cutting or mowing, since grazing involves tread and urine/dung deposition. Lines 195-196: Please revise this title. Line 201 and line 235: The monthly mean temperature was 9.7 °C in August, but the slurry was incubated under 28 °C. The incubation temperature is nearly two times greater than the mean temperature. How would this artificial effect modulate the responses of the measured indices? Line 203: What "them" stands for? Line 220: nitrification again?

3. Results and Discussion

Lines 286-291: I can't see how you can jump from nitrification or denitrification potentials to assessing the contributions of bacterial and fungi to potential $N_2O$ emissions. Nitrification or denitrification potentials should not be regarded as $N_2O$ productions especially emissions by nitrification or denitrification. From this sense, the discussion section should be rewritten thoroughly.

---

## Referee Comment (RC5) · Anonymous Referee #4 · 10 Apr 2018

The present manuscript, entitled "Fungi regulate response of N2O production to warming and grazing in a Tibetan grassland" was interesting. However, there are some critical issues, which may need to be addressed -

(1) The statistical analysis and reporting are weak. Is there any real field replication, excluding any pseudo replication? What was the power of the statistical test? Statistical differences among different treatments were not reported for all the sub-plots. Additionally along with p values, standard Error of the mean difference may need to be reported in the plots to understand the differences between the treatment means better.

(2) It was not clear how were the relative contributions of bacteria and fungi in nitrification, denitrification and total N2O production derived from the total respective

measurements? The methods need to be clear and reproducible.

(3) In addition to the present results of the relative contribution of bacteria and fungi in nitrification and denitrification, the definite mechanisms for bacterial and fungal pathways of nitrification and denitrification need to present to demonstrate the change in the pathway of N2O production under the warming treatment. A definite mechanism of shifting in the relative contribution of bacteria and fungi in N2O production would help the reader to understand the present results in a systematic way, particularly under the warming treatment. This would also help to explain and understand the underline reasons of changing the pathway of N2O production between bacteria and fungi under warming.

(4) It was also not clear why the effects of warming on relative contribution of bacteria and fungi on nitrification, denitrification were diluted when warming treatment was combined with grazing, for example in fig 5 ?

---

## Author Comment (AC1) · 10 Apr 2018

Thank you for you suggestions. We have revised our manuscript "Fungi regulate response of N2O production to warming and grazing in a Tibetan grassland", based on your comments. We have carefully addressed each comment and our responses to these comments are listed the below. The attachments are the manuscript which had improved as your suggestions. We hope that all necessary revisions have been made. However, we would be prepared to make further revisions and modifications if required.

Responses to the Reviewer's comments:

[Comment]- This is a concise and nicely written paper, focusing on fungal and bacterial contributions to potential N2O emissions in an alpine grassland in response to warming

and grazing treatments in the field. The authors report several interesting observations, including an increased bacterial enzyme activity and a decreased fungal enzyme activity for N2O emissions under warming. The results have immediate implications for GHG emissions under the scenario of climate change. I have several suggestions for the authors to consider in order to improve the manuscript.

1. Although the authors showed that fungal and bacterial pathways for N2O emissions changed in different directions under warming, the underlying mechanisms or causes remain unknown. In Line 321-322, it is mentioned that increased NO3–N may inhibit fungal growth. Can you elaborate more? Also, did warming affect soil moisture contents or dynamics compared to the control? If so, how would moisture change affect fungal versus bacterial communities? In the end, I am interested in the driving force leading to the observed changes, it is direct warming effect or indirect effect mediated by other factors? Unless we know answers to these questions, we can hardly speculate on the future changes.

[Responses]- We thank the reviewer for the kind suggestion.

For "fungal and bacterial pathways for N2O emissions changed in different directions under warming, the underlying mechanisms or causes remain unknown." It is the two reasons that lead to the changes of fungal and bacterial pathways for N2O emissions by warming. Firstly, the increased of soil temperature directly reduce fungal activity but increase bacterial activity, because fungi prefer the cold environment compared with bacteria. Secondly, warming indirectly reduce fungal activity but increase bacterial activity through increased soil inorganic N and decreased soil organic N in our site, please see Lines 350-355, because fungi prefer higher organic C/N environment while bacteria prefer higher inorganic C/N environment. All these changes caused the fungal and bacterial pathways for N2O emissions changed in different directions under warming. We have improved the manuscript and make sure the underlying mechanisms is clearly, please see Lines 343-357.

For "In Line 321-322, it is mentioned that increased NO3–N may inhibit fungal growth. Can you elaborate more?". We showed more data to support our findings, at our site, not only the soil inorganic N was increased, as reflected by soil NO3—N concentration (Fig. 2a and 2b); but also the soil dissolved organic nitrogen was significantly decreased from 48 to 41 mg kg-1 (P<0.04). Moreover the soil labile C and N was also found significantly decreased by warming(Rui et al., 2012). Warming indirectly reduce fungal activity but increase bacterial activity through increased soil inorganic N and decreased soil organic N in our site, please see Lines 349-355.

For "did warming affect soil moisture contents or dynamics compared to the control? If so, how would moisture change affect fungal versus bacterial communities?". Yes, warming significantly decreased soil moisture at our site (Fig. 1), but we do not think warming affected fungal versus bacterial communities through the soil moisture. Although the fungi prefer the relative dry soil condition, the NEA and DEA from fungi were not increased, while the NEA and DEA from bacteria were not increased in the warming treatment. This might be duo to the fact that warming induced changes in in soil moisture is not great enough to affect the fungal and bacterial community.

For "I am interested in the driving force leading to the observed changes, it is direct warming effect or indirect effect mediated by other factors?" We believe that warming directly affected the fungal versus bacterial communities due to the increase of the temperature. Additionally, warming also indirectly mediated the fungal versus bacterial communities through the changes in the substrate. We had dicussed it in the first section, improved the manuscript and make sure the underlying mechanisms is clearly, please see Lines 343-357.

[Comment]- 2. Speaking of future predictions, I think it should be emphasized that measurements made here were potential rather than "real" emissions in the field. A critical requirement for denitrification to occur is anoxic or sub-oxic conditions. Therefore, I would think that N2O emissions more depend on the hydrological or redox conditions of the soil. Observations of fungal and bacterial enzyme activity changes in the lab may or may not apply to the field observations, depending on how warming affects soil moisture.

[Responses]-For "Speaking of future predictions, I think it should be emphasized that measurements made here were potential rather than "real" emissions in the field." We fully agree with the referee that the fungal and bacterial enzyme activities cannot be shown as the result of N2O emissions. The measurements under laboratory incubation reflected the potential ability of the soil fungal and bacterial activities in nitrification and denitrification because such laboratory incubation could avoid the impacts of various confounding factors and well clarify the mechanism responsible for N2O emission. At revised version, we clarified that our measurements in the laboratory indicated the potential emission.

For "A critical requirement for denitrification to occur is anoxic or sub-oxic conditions. Therefore, I would think that N2O emissions more depend on the hydrological or redox conditions of the soil." Yes, we also fully agree with the referee that anoxic or sub-oxic conditions and soil moisture is very important for N2O emissions. For hydrological or redox conditions, because we did not measure it, so it is hard to dicussed it directly, but it was mainly influenced by soil moisture, The soil moisture was showed in Fig. 1c.

For "Observations of fungal and bacterial enzyme activity changes in the lab may or may not apply to the field observations, depending on how warming affects soil moisture." The observations of fungal and bacterial enzyme activities were also not appled as the field emissions, they were used to clarify the mechanism responsible for N2O emission. In our stie, the filed N2O emission in 2011-2012 was shown in the manuscript. And the laboratory measurements of the total nitrification and denitrification enzyme activities all were the same with the filed N2O emission at our site (Zhu et al. 2015; Fig. 4c and 4f), which showed it could well clarify the mechanism responsible for N2O emission.

For "depending on how warming affects soil moisture". Although warming significantly decreased the soil moisture at our site, the field N2O emission, total nitrification and denitrification enzyme activity did not change as a result of warming (Zhu et al. 2015; Fig. 4c and 4f). It might be due to the fact that the changes in soil moisture by warming was not great enough to lead to a detectable difference in field N2O emission, total nitrification and denitrification enzyme activity.

[Comment]-Some minor points: Line 163: I notice that there was no field replicate for the measurement?

[Responses]- In this study, we used in field replicates. There were four replicates for each of four treatments. Therefore, twe had 16 plots in total. We collected soil samples from each plot. We made detailed description on how to collect soil in the revised version, please see lines 139-143.

[Comment]-Line 223: N2 not N.

[Responses]- Corrected

[Comment]-Line 227: Why only three time points for the denitrification measurement versus 5 points for nitrification?

[Responses]-For DEA incubation experiment, we collected at least 12 ml gas for N2O concentration measuring. If too many times were used to collect N2O, it would change the incubation pressure and influence the responsibility of the experiment. So, we only collected 3 times in the incubation experiment. But for NEA incubation experiment, it does not matter. Additionally, different sampling times for NEA and DEA should have little effect on the reliability of our results because this study did not aim to distinguish the contribution of total nitrification and denitrification to N2O emissions. Here we just estimated nitrification enzyme activity by analyzing the change of NO2-+NO3- concentration after incubation, see lines 212-214 and denitrification enzyme activity by analyzing the change of N2O concentration after incubation, see lines 236-240. Overall. we only compared NEA and DEA among all treatments, respectively.

[Comment]-Lines 285 and 292: NEA, DEA, FDEA, BDEA: : :not used in the previous text.

[Responses]-For NEA, DEA, we changed it to TNEA and TDEA. They were used in the previous version, please see lines 279-288.

[Comment]-Line 304: I don't think IC is much higher in Haibei soils than some temperate grassland soils in Mongolia. IC contents are dependent on soil pHs. . . [Responses]-Thank you for your suggestion. We corrected it in the new version.

**Supplement:**

**Fungi regulate response of N₂O production to warming and grazing in a Tibetan grassland**

[revised manuscript text omitted]

PCR performed as follows: 20 s at 95 °C, 30 s at 62 °C and 30 s at 72 °C. A final 5-min extension step completed the protocol.

**2.4 Potential total nitrification/denitrification enzyme activity, and fungal and bacterial nitrification/denitrification enzyme activity analysis.**

[revised manuscript text omitted]

Soil temperature varied from 11.8 to 14.0 ℃. Grazing (P=0.05) and warming (P<0.01)

increased soil temperature. The average soil moisture varied from 26% to 34% (w/w).

Grazing had no effect on soil moisture, which was lower in warming plots (P<0.01)

(Fig. 1c). There was an interactive effect between grazing and warming on soil temperature (P<0.01).

Soil total C (TC) was not affected by grazing (P=0.13) or warming (P=0.12) alone, but there was a marginal interaction between grazing and warming on TC (P=0.07) (Fig.

2a). Similar to TC, soil total N (TN) also showed no response to grazing or warming (Fig. 2b). Soil $NH_4^+$-N content was lower in warming treatments than in no-warming treatments (P=0.05) (Fig. 2c). Greater soil $NO_3$-N content occurred under the warming treatments (P=0.05) than under the no-warming treatments (Fig. 2d).

**3.2 Microbial functional genes**

Bacterial gene abundance varied from $4.71 \times 10^9$ to $5.93 \times 10^9$ copies $g^{-1}$ dry soil, which was much higher than fungal gene abundance (Fig. 3). Warming and grazing both increased the bacterial gene abundance in soils (P<0.01), but there was no interaction effect between them on bacterial gene abundance. By comparison, fungal gene abundance showed no difference across all treatments.

**3.3 Nitrification and denitrification enzyme activity from bacteria and fungi**

Total nitrification enzyme activity (TNEA) varied from 1.07 to 1.64 μg N $g^{-1}$ $h^{-1}$ in all treatments. Bacterial nitrification enzyme activity (BNEA) ranged from 0.43 to 0.64

μg $g^{-1}$ $h^{-1}$, which was lower than the fungal nitrification enzyme activity (FNEA) in soils (0.59–0.66 µg g$^{-1}$ h$^{-1}$) (P=0.01) (Fig.4 a-c). FNEA was lower under warming treatments than under the no-warming treatments (P=0.05).

Total denitrification enzyme activity (TDEA) was between 1.32 and 1.80 µg N g$^{-1}$ h$^{-1}$

$^{1}$. Fungal denitrification enzyme activity (FDEA) was clearly the dominant process for

TDEA (Fig. 4 d-f), because it was higher than bacterial denitrification enzyme activity (BDEA) for all treatments except warming. Warming increased BDEA (P=0.04).

Warming and grazing had a significant interaction effect on FDEA (P<0.01).

**3.4 The contribution of bacteria and fungi to potential N$_2$O emissions**

The contribution of FNEA to TNEA varied from 47% to 56%, and the contribution of FDEA to TDEA varied from 45% to 63% (Fig. 5). Warming significantly decreased the contribution of FNEA and FDEA to TNEA and TDEA in soils (FNEA: P=0.02;

FDEA: P=0.04). There were no differences in the contribution of FNEA and FDEA to

TNEA and TDEA in any treatments.

**4 Discussion**

N$_2$O was mainly produced from the microbial nitrification and denitrification processes, but the microbial pathway of these processes was still unclear. In our results, fungi contributed 54% and 63% of the TNEA and TDEA, respectively, in the alpine grassland studied. Our result of the fungal contribution to N$_2$O production is much lower than Laughlin and Stevens (2002) and Zhong *et al.* (2018) who reported 89% and

86% fungal contribution from temperate grasslands, but is higher than the 40-51%

fungal contribution observed across different ecosystems by Chen et al. (2014). Kato et al. (2013) showed that N$_2$O emissions from FDEA was higher than from BDEA in alpine meadows, reinforcing the important role fungi play in the N$_2$O production process. Our findings support our first hypothesis and further proved that both denitrification and nitrification were largely driven by fungal communities in alpine grasslands. A possible explanation is that fungi prefer the arid, high complex compounds subtract substrate and low-temperature environment (Pietikäinen et al.,

2005;Chen et al., 2015;Marusenko et al., 2013). In alpine grasslands, the mean annual temperature is 0 $^{\circ}$C; even during the sampling day the mean temperature was only 11

$^{\circ}$C. The cold environment could cause higher activity in fungi than in bacteria.

Moreover, the cold environment decreases the rate of mineralization, leading to greater

C and N accumulation (Ineson et al., 1998;Schmidt et al., 2004). In our study, soil TC

and TN concentrations were 72–86 g kg$^{-1}$ and 6–7 g kg$^{-1}$, respectively (Fig. 2a and 2b), much higher than in temperate grasslands and farmland, providing a favorable environment for fungi (Bai et al., 2010). Inorganic C and inorganic N content were also much higher than study in temperate grasslands (Zhong et al. 2018), but lower than temperate farmland ecosystems (Chen et al., 2015;Laughlin and Stevens, 2002); this is mainly because the fungal contribution to N$_2$O potential and N loss in the alpine grasslands was lower than in temperate grasslands but higher than farmland on the

Qinghai-Tibetan Plateau.

Our methodology did not exclude a role for archaea in nitrification and denitrification. Previous studies on grasslands only focused on fungal and bacterial process because archaeal specific inhibitors have not yet been identified for N cycling processes. However, archaea are widespread in soils, are involved in nitrification denitrification (Cabello et al., 2004), eg. archaeal ammonia oxidizers are common globally (Leininger et al., 2006). In our study, we also found the TNEA was higher than the sum of NEA from bacteria and fungi, while TDNA was higher than DEA from bacteria and fungi (Fig. 4), which showed that archaea also played the role in $N_2O$

producing process in our site. However, it included the archaeal and abiotic components.

The development of inhibitor-based approaches may help to show how archaea responses to environmental change (Marusenko et al. 2013).

Our results supported our second hypothesis that although warming did not change the potential $N_2O$ emissions on the Qinghai-Tibetan Plateau, the biotic pathways responsible for $N_2O$ had been changed, as bacterial contribution to $N_2O$ potential was higher than fungal under the warming treatment (Fig. 4). The increase in bacterial $N_2O$

production potential, coupled with decrease in fungal $N_2O$ production, could be the main reason why there was no difference between control and warming treatments. The field data in our site was measured in year of 2011–2012 and also showed no effect of warming on $N_2O$ emission (Zhu et al. 2015). Our results reinforced this and suggested that bacterial nitrification and denitrification alone is unable to accurately describe the response of $N_2O$ to warming. It is the two reasons that lead to the changes of fungal and bacterial pathways for $N_2O$ emissions by warming. Firstly, the increased of soil temperature directly reduce fungal activity but increase bacterial activity, because fungi prefer the low-temperature environment environment compared with bacteria.

Secondly, warming indirectly reduce fungal activity but increase bacterial activity through increased soil inorganic N and decreased soil organic N, because fungi prefer higher organic C/N environment while bacteria prefer higher inorganic C/N environment. In our site, although the soil $NH_4^+$-N concentration did not change with warming, soil $NO_3^-$-N concentration was significantly increased showed the soil inorganic N was increased (Fig. 2a and 2b); on the other hand, the soil dissolved organic nitrogen was significantly decreased from 48 to 41 mg kg$^{-1}$ (P<0.04), the soil labile C and N was also found significantly decreased by warming (Rui et al., 2012), it showed the soil organic C and N was decreased in our site. All these changes could directly and indirectly inhibit the growth of fungal communities and their activity, but increase those of bacteria. Although the gene abundance of fungi was not changed, the FNEA and FDEA were reduced by 16% and 30% respectively by warming, and BDEA was increased by 41%. All these changes resulted in fungi contributing less to nitrification and denitrification than bacteria (Fig.5). This indicates that the soil microbial process was altered by warming, even though the TNEA and TDEA did not change, with a shift in the dominance from fungi to bacteria on $N_2O$ production after 10 years of warming.

Numerous studies have demonstrated that grazing can impact microbial processes and induce the loss of N through: (1) altering the substrate concentration for $N_2O$ production and reduction in soil through the deposition of dung and urine (Saggar et al., 2004); (2) reducing vegetation cover due to changes in soil water content and energy balance (Leriche et al., 2001); and (3) increasing soil compaction and reducing soil aeration through animal tramping (Houlbrooke et al., 2008). However, in this study fungal and bacterial nitrification and denitrification activity showed little response to winter grazing. A possible explanation is that neither soil moisture, plant biomass nor organic/inorganic C/N content were affected by winter grazing (Fig.1-2). Additionally, the soil was frozen in winter, so that the effect of selective feeding and trampling could be limited by grazing sheep rather than other livestock (Zhu et al., 2015; Krümmelbein et al., 2009). As a result, the same soil environmental conditions for both winter grazing and control had no effect on soil fungi and bacteria, and thus on fungal and bacterial nitrification and denitrification. Moreover, the field data of $N_2O$ emission in the year of

2011-2012 also supports the results of Zhu et al. (2015) and suggests that replacing summer grazing by winter grazing could cause the soil N cycle process to become stable.

Overall, we conclude that fungi played the dominant role in the soil N cycle, and could be the major source of $N_2O$ production and N loss in alpine meadows. Climate warming is not likely to affect potential $N_2O$ emissions but 
[revised manuscript text omitted]

denitrification enzyme activity (BDEA) (d), fungal denitrification enzyme activity
(FDEA) (e) and total denitrification enzyme activity (TDEA) (f) in an alpine meadow.
C(■), control treatment; G (□), winter grazing treatment; W (▨), warming treatment;
WG (▥), warming combined with the winter grazing treatment. Values are means ±1
s.e.m. (*n*=4). Different letters indicate significant differences within each treatment
(P<0.05).

**Fig. 5** Contribution of bacteria and fungi to total nitrification enzyme activity (box with
the red and dashed line) and total denitrification enzyme activity (box with the black
and solid line) in an alpine meadow. C(■), control treatment; G (□), winter grazing
treatment; W (▨), warming treatment; WG (▥), warming combined with the winter
grazing treatment. Values are means ±1 s.e.m. (*n*=4).

Fig.1

[Figure]

Fig. 2

[Figure]

Fig. 3

[Figure]

Fig.4

[Figure]

Fig.5

[Figure]

---

## Author Comment (AC2) · 10 Apr 2018

Thank you for you suggestions. We have revised our manuscript "Fungi regulate response of N2O production to warming and grazing in a Tibetan grassland", based on your comments. We have carefully addressed each comment and our responses to these comments are listed the below. The attachments are the manuscript which had improved as your suggestions. We hope that all necessary revisions have been made. However, we would be prepared to make further revisions and modifications if required.

[Comment]- This study reports the effect of warming and grazing on soil biotic contribution to N2O production in a Tibetan grassland, by examining a long-term (over 10 years) experiment combined with an incubation experiment. Their results indicated

that fungi could be the main source for N2O production potential in the Tibetan alpine grasslands. Overall, the manuscript is of interest and generally well written. But there are some concerns and unclear points that should be addressed prior to publication. Please find some more detailed comments below.

1. Lines 162-164, is it enough to collect only 5 cores for each soils?

For soil sampling, "randomly collected" was used to reduce the spatial heterogeneity. At our site, soil samples were collected using this method in all related experiments because the plot area used for warming was limited. However, this sampling method was proved to be suitable which can be found in a series of our published papers in Ecology (Wang et al. 2012), Global change biology (Luo et al. 2010), Journal of soils and sediments (Rui et al. 2012) and so on.

2. How the authors draw the contribution of bacteria and fungi to total nitrification enzyme activity and total denitrification enzyme activity as shown in Fig. 5? I cannot find the specific description in the section "Materials and Methods".

Thank you for your comment. In the new version, we added the description in Materials and Methods". Please see the lines 245-248.

3. Line 257, "Fig. 1A" should be changed to "Fig. 1a", based on the Figure 1. Also, the authors should revised it throughout the main text.

Corrected.

4. Line 259, "soil moisture" should be changed to "The average soil moisture".

Corrected.

5. In Figs. 1 and 4, why significant differences were only shown in Figs. 1b and 4e rather than all of subfigures?

We have removed different letters from the Figs. 1b and 4e. The two-way ANOVA results in all figures were enough. Please see lines 589-611.

References

Luo, C., Xu, G., Chao, Z., Wang, S., Lin, X., Hu, Y., Zhang, Z., Duan, J., Chang, X., and Su, A.: Effect of warming and grazing on litter mass loss and temperature sensitivity of litter and dung mass loss on the Tibetan plateau, Global Change Biol., 16, 1606-1617, 2010. Rui, Y., Wang, Y., Chen, C., Zhou, X., Wang, S., Xu, Z., Duan, J., Kang, X., Lu, S., and Luo, C.: Warming and grazing increase mineralization of organic P in an alpine meadow ecosystem of Qinghai-Tibet Plateau, China, Plant Soil, 357, 73-87, 2012. Wang, S., Duan, J., Xu, G., Wang, Y., Zhang, Z., Rui, Y., Luo, C., Xu, B., Zhu, X., and Chang, X.: Effects of warming and grazing on soil N availability, species composition, and ANPP in an alpine meadow, Ecology, 93, 2365-2376, 2012.

Please also note the supplement to this comment:
https://www.biogeosciences-discuss.net/bg-2017-552/bg-2017-552-AC2-supplement.pdf

―――――――――――――――

---

## Referee Comment (RC6) · Anonymous Referee #6 · 13 Apr 2018

The paper presented an interesting topic, which focused on fungi regulating the responses of N2O production to warming and grazing treatments in Tibetan grassland. The authors report several new information, such as an increased bacterial enzyme activity and a decreased fungal enzyme activity of regulating N2O emissions under warming treatment. The findings have implications for well-understanding the responses of N2O emissions to the scenario of climate change and/or disturbance. However, there are sevel concerns need to be addressed.

1. The description of experimental desigh is not clear, particularly, there is a confusing in introducing winter grazing treatment. What is the reason for the selection of winter grazing treatment in present study? Tibetan grassland is experienced to be covered by snow, frozen soils, and the grass should be withered in winter. In the same plots,

the ecological effects of winter grozing should be interferenced by previous different grazing treatments (lines 153-156). How to avaid it?

2. Potential total nitrification/denitrification for N2O emission rate from incubation experiment is not a "real" rate of N2O emission under the field conditions. In terrestrial ecosystems, soil temperature, moisture, pH, soil N availability, and DOC etc. are generally considered as the major factors of controlling N2O emissions. For this study, the lack of field simultaneous monitoring data of N2O rates is a critical issue. Although the authors tried to cite the previous results for discussion, the conclusion obtained from an incubation experiment is still not general acceptable.

3. The underlying mechanisms that fungal and bacterial pathways for controlling N2O emissions remain unkonwn. The authors need to elaborate the relative contributions of fungi and bacteria in nitrification and denitrification processes of N2O productions.

4. Line 130-131: The symbol oC is not correct.

5. There are several mistakes in English writing, which should be revised throughout the text.

BGD

---

## Author Comment (AC3) · 16 Apr 2018

Thank you for you suggestions. We have revised our manuscript "Fungi regulate response of N2O production to warming and grazing in a Tibetan grassland", based on your comments. We have carefully addressed each comment and our responses to these comments are listed the below. The attachments are the manuscript which had improved as your suggestions. We hope that all necessary revisions have been made. However, we would be prepared to make further revisions and modifications if required.

Responses to the Reviewer's comments:

[Comments] This manuscript presents an interesting study on the response of an alpine grassland ecosystem to warming and grazing in the period of 10 years. N2O produc-

tion via variable microbial components was the main focus. It is written concisely and easy to understand. However, regarding the experiment design and interpretation of the data set, I believe that there is still more to improve before it could be published. Despite their investigation into multiple treatments and parameters, the authors need to provide more field evidence and literature comparison to reach a convincing conclusion. Throughout the whole manuscript, the authors seem to mix up denitrification enzymatic activity and N2O production. If the inhibitors applied in the experiments to determine denitrification rates also inhibit N2O reduction to N2, the N2O production should rather represent potential denitrification rates. If N2O reduction was not inhibited during the experiment, the results could not be noted as "denitrification rates". Please clarify this key point and make revision accordingly. The methods determining these rates should be described in more details in M&M.

[Responses] Based on the referee #1 's suggestions, we provided more field data and literature to support our conclusion. The filed N2O emission in 2011-2012 at our site (Zhu et al. 2015) was referenced in our manuscript, please see lines 338-339 and lines 378-379. We also added the mean temperature and rainfall data during the sampling year and months; the soil dissolved organic nitrogen data in our manuscript, please see lines 130-132 and lines 346-349. Because these data were obtained by other colleagues, we cannot present them as figures in the current study. The filed N2O emission supported our conclusion of warming had no effect on total nitrification and potential of N2O production from denitrification. The soil dissolved organic nitrogen data supported our conclusion of warming reduced the potential of N2O from fungi because of the reduction of organic substrates. We also showed more references to supports our conclusions, e.g. Zhu et al. (2015) to support our conclusion of warming had no effect on total nitrification and potential of N2O production from denitrification; the results of Zhu et al. (2015), Krümmelbein et al. (2009) and Steffens et al. (2008) supported our conclusion of winter grazing had little effect on environment because the soil is frozen in winter and often covered with snow and grazing has little effect on soil conditions, please see lines 338-339 and 374. To determine potential denitrification rates, we incubated soil samples under anaerobic condition and did not add any inhibitor to inhibit N2O reduction to N2 process. Therefore, our results only can be presented as the potential of N2O emission from denitrification. We have clarified this in M&M, please see lines 196-251.

[Comments] Line 111: "To clarify whether fungi control the N2O production process" is misleading as Fungi contributes anyway; I assume that the authors wish to clarify the "role of fungi in N2O production process"

[Responses] Done as your suggestion. please see lines 112-113.

[Comments] Line 161-162: Please explain this; why do you see the effects on ecosystem level despite that plot size are 3 m? Any data to support this?

[Responses] This is really good question. The plot size used for warming treatments are generally small, less than 1 m2 (Cantarel et al. 2012) to more than 10 m2 (Long et al. 2015). These studies well showed the effects of treatments on ecosystem (Cantarel et al. 2012; Long et al. 2015). In this study, the size of our plots was considered according to three points: 1) A little big size was used because grazing was involved. Although the size of plot might affect the animal feeding activities, all experimental sheep were fenced into three additional 5*5 m fenced plots for one day before the beginning of the grazing experiment to help them adapt to small plots for reducing the experimental error. 2) The warming efficiency and cost (we used the infrared heaters in warming treatments for increasing soil temperature) was another factor; and 3) the species composition and vegetation coverage is even in this grassland. Previous publications (Wang et al. 2012 Ecology, Luo et al. 2010 Global Change Biology, Luo et al. 2009 Soil Biology and Biochemistry, Rui et al. 2012 Journal of Soils and Sediments) from this study have demonstrated that the plot size can show the effects on ecosystem level.

[Comments] Line 165: If 10 years' warming and grazing treatment was done, why was only one sampling of soils by the end of 10 years' treatment? Have you considered the soil heterogeneity between control and treatment plots since the beginning of treatments?

[Responses] Only one sampling of soils was done by the end of 10 years treatment. The reason is that this is the first time for us to pay attention to the contribution of fungi and bacteria to N2O production based on recent research advances and fresh soil is required for microbial analysis especially for the incubation experiment. A thorough understanding about the long-term impact of warming and grazing on soil fungal nitrification and denitrification from alpine meadow grassland requires further investigation through multi-sampling during a long period. We mentioned this limitation in Discussion, please see lines 383-386. Additionally, we considered the soil heterogeneity between control and treatment plots since the beginning of treatments. There is no difference between treatments the beginning of this experiment. To reduce the soil heterogeneity, all the plots were asigned in a complete randomized block. For "soil heterogeneity between control and treatment plots since the beginning of treatments?". We think the spatial heterogeneity was exit in everywhere.

[Comments] Line 166: Including or excluding organic layer? Please specify.

[Responses] Done as your suggestion. please see lines 167.

[Comments] Line 225-226: 100% of water-holding capacity could favor denitrification; however, it may not likely represent field condition, which is usually drier. Please justify your choice of such incubation condition.

[Responses] The incubation experiment was used to show the potential of N2O produce from denitrification of soil, it cannot be represented as the N2O production of field. The 100% of water-holding capacity was provided an relative good environment for denitrification so that can inspire the activities of denitrifying microorganism and show the ability N2O produce by denitrifying microorganism in soils. The method and the incubation condition was commonly used to measure the denitrification enzyme activity and proved to be useful (Smith and Tiedje, 1979; Simek and Hopkins, 1999;

Chroňáková et al. 2009; Cantarel et al. 2012).

[Comments] Line 294-298: Use present tense: use "is" to replace "was".

[Responses] Done as your suggestion. Please see lines 299-230

[Comments] Line 298: Change "who" to "whom".

[Responses] Done as your suggestion. Please see lines 303

[Comments] Line 314 to 315: When comparing the studied alpine grassland to temperate grassland, how do come to the conclusion that the lower inorganic C and N contents in soil were due to larger fungal contribution to N2O production? What about the higher mineralization rates in the temperate systems? In addition, the control of inorganic C or N levels in soil could be also related to biomass uptake and turnover. Please clarify it and avoid such speculation.

[Responses] We fully agree with the referee that the lower inorganic C and N contents in soil based on observations from alpine grassland to temperate grassland cannot come to the conclusion. In the new version, we removed the sentence and improved this part to avoid such speculation. Please see lines 319-320

[Comments] Line 324: "common" and "globally" do not fit together; please revise.

[Responses] Done as your suggestion. Please see lines 325

[Comments] Line 348-349: "gene abundance of fungi was not changed" against treatments; how do you reconcile your finding with the hypothesis?

[Responses] The gene abundance of fungi was not changed by warming, but warming changed FNEA and FDEA. Such inconsistency between gene abundance of fungi and FNEA/FDEA might be explained by the fungal gene abundance not providing information on real-time process rates. The reason is that process rates are largely dependent on environmental conditions. Fluctuations in environmental conditions can cause rapid changes in real-time process rates, but do not necessarily affect gene abundance
(Zhong et al. 2014). We have improved it in the new version, please see lines 353-358

Overall, we conclude that fungi played the dominant role in the soil N cycle, and could be the major source of $N_2O$ production and N loss in alpine meadows. Climate warming is not likely to affect potential $N_2O$ emissions but 
[revised manuscript text omitted]

[Figure]

Fig. 2

[Figure]

Fig. 3

[Figure]

Fig.4

[Figure]

Fig.5

[Figure]

---

## Short Comment (SC1) · 29 Apr 2018

*A note upfront from the submitting person: This review was prepared by Oliver Vögeli and Ursina Morgenthaler, master students in geography or earth system science at the University of Zurich. The review was part of an exercise during a second semester master level seminar on "the biogeochemistry of plant-soil systems in a changing world", which I organize. We would like to highlight that the depth of scientific knowledge and technical understanding of these reviewers represents that of master students. We enjoyed discussing the manuscript in the seminar, and hope that our comments will be helpful for the authors.*

The aim of Zhong et al. (2018) is to clarify the role fungi play in the loss of N2 and

climate warming via N2O production in an alpine meadow. Their investigation concentrates on changes in climate and land use management. For this, they examined the effects of 10 years' warming and winter grazing on soil N2O emissions potential in an alpine meadow on the Tibetan plateau. Zhong et al. (2018) found, that warming and winter grazing had no effect on overall nitrification and denitrification. Warming changed only the biotic pathways from fungi domination to bacterial domination, but did not change total nitrification or denitrification. This study is important, because previous studies in Tibetan alpine grasslands mainly focused on bacterial nitrification and denitrification processes. The findings of Zhong et al. (2018) thus open new possible scenarios, which can refine greenhouse gas flux models.

The paper brings new interesting observations about the role of fungi in N2O production in alpine grasslands. The introduction is very well and comprehensibly written and gives a complete overview on the study. The duration of the study (10 years) seems appropriate to measure microbial behaviour. The authors did an extensive laboratory examination, where they measured many soil proper-ties (soil moisture, soil mineral N, total C and N content, soil DNA). In the discussion, the authors also consider the impact of archaea (line 319f), which are often forgotten when consider microbes.

The methods used seem appropriate in general, however some questions arise with regards to measurements and sampling. The paper leaves open why the difference was set to 1.2°C and 1.7°C during day and night respectively in summer. Furthermore, it is not clear what the effect of 1500 W are in winter. The authors mention that some thermometers are broken, but it would have been nice to get at least the data from the working thermometers. Zhong et al. (2018) mention from the begin-ning and in the title, that the effect of winter grazing was under investigation. However, for half of the time there was summer grazing on the sites. Please describe this treatment further.

Questions: General: Samples were taken on one only day. Would it be possible, that due to special environmen-tal circumstances on that day, the results were in some way not representative? The authors do not explain why they chose to simulate winter

grazing and not summer grazing. If it is the reason mentioned in line 370, the authors should explain it already in the introduction. Consider using less acronyms, it is sometimes hard to follow the story. Rethink if 'treatment' (e.g. summer grazing treatment) really needs to be used that often. In the introduction: Maybe elaborate more on the state of art and on similar studies done in other parts of the world.

259: What does (w/w) mean?

290: What does the sentence mean? There were no differences in the contribution of FNEA and FDEA to TNEA and TDEA in any treatments. There are differences, aren't there? 306: What does 'high complex compound substract substrate' mean?

336/369: Please elaborate on 'field data from 2011-2012'? It is not clear to us what this refers to.

341: We do not understand the sentence starting with "In our site...". Maybe you can clari-fy/reformulate that.

364f: We do not understand the sentence starting with "Additionally...". Please elaborate on why the effect of sheep is limited compared to other livestock?

Figure 1 and 4: The distribution of the letters indicating the significant differences is inconsistent, why do you only show it in section b of figure 1 and in section e of figure 4? Also think about using other symbols, since these letters might be confused with the letters for the figure subdivision.

Figure 5: From this figure we read that fungi and bacteria come from the hard rock substrate, that the denitrification happens in the subsoil and the nitrification in the topsoil. Is that right? Furthermore, we do not understand why W and WG are yellow shadowed and why 'bacteria' is written in purple, while the arrow is green. Also, it is not necessary to make the figure in 3D.

Typos/ remarks concerning structure: 54f: 'Potential' is used too many times. 153: This sentence is formulated rather complicated, maybe you can split it in two sentences. The

term 'grazing treatments' is repeated a lot in those lines, maybe you can replace it? 220: Denitrification enzyme activity 278: forgot N in unit 279: forgot N in unit 316-318: Does this conclusion not contradict to line 103? 322: nitrification and denitrification. 334: fungal N2O production potential.

―――――――――――――――――――

---

## Author Comment (AC4) · 30 Apr 2018

Thank you for you suggestions. We have revised our manuscript "Fungi regulate response of N2O production to warming and grazing in a Tibetan grassland", based on your comments. We have carefully addressed each comment and our responses to these comments are listed the below. The attachments are the manuscript which had improved as your suggestions. We hope that all necessary revisions have been made. However, we would be prepared to make further revisions and modifications if required.

Responses to the Reviewer's comments:

[Comments] 1. The experimental design is not acceptable. Firstly, why did you choose "winter grazing"? There seems no explanation. The temperature should be too low to let the animal grazing out of the field in winter. Additionally, the grassland is expected to be covered by snow and the grasses should be withered in winter. Secondly, the description of the treatment is really confusing. Winter grazing should be used in the current study, but "For grazing treatments, the grazing treatments in this site were used for summer grazing treatments until 2010, from 2011 to 2015, there was no grazing during the summer, and grazing was replaced by cutting and removing about 50

[Responses] Sorry, our previous description caused the misunderstanding by the referee. In the new version, we clarified why we used winter grazing. On the Qinghai-Tibet plateau, winter grazing is very commonly and alpine meadows are generally classified into two grazing seasons, i.e. warm season grazing from June to September and cold season grazing from October to May even the grassland was covered by snow (Cui et al., 2015). Winter pasture contributed about 40

In the new version, we clarified our design. During 2006-2010 summer grazing treatments was used to explore the effects of warming and grazing on ecosystem during the growing seasons (Luo et al. 2010; Hu et al. 2010; Wang et al. 2012). Considering strong disturbance, grazing was stopped during 2011-2015. Given the importance of winter grazing, winter grazing during the non-growing seasons was further investigated (Zhu et al. 2015; Che et al. 2018). We agree with the referee that grazing cannot be simulated by cutting or mowing since grazing involves tread and urine/dung deposition. However, during winter, such effects could be very small because soil and dung are frozen and tread has little effect on soil. Actually, we had examined how clipping simulated the effects of actual grazing before we established four replicated "actual grazing treatments" compare with the "simulated grazing treatments", the soil and plant all showed no difference between simulated grazing and actual grazing treatments (Klein et al. 2004; 2007), and showed the urine/dung deposition and tread by animals' effect on soil and plant is limited. We believe that removal of litter can stimulate the effect of winter grazing, which has been demonstrated by previous studies (Zhu et al. 2015; Che et al. 2018). We had improved the description of the winter grazing treatment and make it more clearly, please see lines 159-174.

[Comments] 2. I can't see how you can jump from nitrification or denitrification potentials to assessing the contributions of bacterial and fungi to potential N2O emissions. Nitrification or denitrification potentials should not be regarded as N2O productions especially emissions by nitrification or denitrification. From this sense, the discussion section should be rewritten thoroughly.

[Responses] Most studies mainly focused on the contribution of bacterial nitrification and denitrification to potential N2O emissions. Because numerous studies have shown that fungal nitrification and denitrification can play an important role in N2O production. Therefore, in this study we aimed to quantify the contribution of fungal and bacterial to potentials of N2O from nitrification and denitrification. Because the contribution of fungal nitrification and denitrification was higher than bacteria's (Fig. 5) in control treatment, this indicates that the fungi played the major role in potential N2O emissions.

We agreed with the referee that the nitrification or denitrification potentials should not be regarded as N2O productions especially emissions. In this study, we mainly focused on the mechanism of N2O produce process and distinguished the role of bacteria and fungi in N2O produce process. In the new version, we rewrote the discussion section and related sections to avoid the misunderstanding.

[Comments] 3. The manuscript is not well prepared. There are lots of writing issues throughout the manuscript. I only presented few of them since there are too many.

[Response] In the new version, we almost rewrote the manuscript and asked a native English speaker Miss Ri Weal to polish the language errors. We hope the new version is easy to read and follow.

[Comments] Abstract Lines 44-46: The treatments should be described briefly in the abstract to increase the readability. Additionally, some key information about the method should be presented.

[Responses] Done. Please see lines 44-48.

[Comments] Lines 46-52: The values should be presented with uncertainties, e.g., standard error, standard deviation or 95

[Responses] Done. Please see lines 48-49 and Fig.5.

[Comments] Lines 46-47: Were these values got from the control?

[Responses] Yes, these values are obtained from the control. We clarified this in the new version, please see lines 49.

[Comments] Lines 49-52: Suggest rephrase these sentences in such way: "However, warming significantly increased the enzyme activity of bacterial nitrification and denitrification to 53

[Responses] Done. Please see lines 53-56.

[Comments] Lines 52-54: How could you make this conclusion? Under what conditions do soil fungi contribute more to N2O production? This sentence is of course not clear. If the conclusion is obtained based on results from the control, it should be put somewhere after lines 46-47. Additionally, can you make such a strong conclusion based on an incubation experiment?

[Responses] Thank the referee for pointing out the question. We rewrote the abstract as the referee suggested, please see lines 40-62. Our conclusion was based on the role of fungi and bacteria in N2O produce process by the incubation experiment but not in N2O emissions. In the new version, we clarified this, please see lines 1-62.

[Comments] Lines 56-58: This should not be put in the abstract as a key implication since it should be regarded as a fact.

[Responses] Done. Please see lines 58-60.

[Comments] Line 59-60: This sentence should be rephrased since some grammar issue exists. For example, "lead to refine: : :." is not correct. Overall, the abstract needs substantial revision.

[Responses] Done. Please see lines 40-62.

[Comments] Introduction Line 66: not clear what does "it" refer to.

[Responses] "it" refer to N2O emission, we clarified it, please see lines 67-69.

[Comments] Lines 67-69: This sentence needs substantial revision.

[Responses] Done, please see lines 69-71.

[Comments] Line 122: Why did you choose "winter grazing"? There seems no explanation. The temperature should be too low to let the animal grazing out of the field in winter. Additionally, the grassland is expected to be covered by snow and the grasses should be withered in winter.

[Responses] Sorry, our previous description caused the misunderstanding by the referee. In the new version, we clarified why we used winter grazing. On the Qinghai-Tibet plateau, winter grazing is very commonly and alpine meadows are generally classified into two grazing seasons, i.e. warm season grazing from June to September and cold season grazing from October to May even the grassland was covered by snow (Cui et al., 2015). Winter pasture contributed about 40

[Comments] M M Lines 130-131: The symbol C is not correctly used.

[Responses] Done, please see lines 136-137.

[Comments] Lines 131-132: over 80

[Responses] Over 80

[Comments] Lines 133-134: Please clearly present the soil classification systems and the references.

[Responses] Done, please see lines 139.

[Comments] Lines 134: There should be a space between the word and the parentheses here and in other sentences or Figures (Please check the figures as well).

[Responses] Done, please see lines 142 and the caption of figures.

[Comments] Line 139: The indent here is not consistent with other paragraphs. Please keep consistency.

[Responses] Done, please see lines 145.

[Comments] Line 146: delete was.

[Responses] Done, please see lines 152.

[Comments] Lines 153-156: The description is really confusing. According to the above paragraph, winter grazing was used in the current study, but "For grazing treatments, the grazing treatments in this site were used for summer grazing treatments until 2010, from 2011 to 2015, there was no grazing during the summer, and grazing was replaced by cutting and removing about 50

[Responses] In the new version, we clarified our design. During 2006-2010 summer grazing treatments was used to explore the effects of warming and grazing on ecosystem during the growing seasons (Luo et al. 2010; Hu et al. 2010; Wang et al. 2012). Considering strong disturbance, grazing was stopped during 2011-2015. Given the importance of winter grazing, winter grazing during the non-growing seasons was further investigated (Zhu et al. 2015; Che et al. 2018). We agree with the referee that grazing cannot be simulated by cutting or mowing since grazing involves tread and urine/dung deposition. However, during winter, such effects could be very small because soil and dung are frozen and tread has little effect on soil. Actually, we had examined how clipping simulated the effects of actual grazing before we established four replicated "actual grazing treatments" compare with the "simulated grazing treatments", the soil and plant all showed no difference between simulated grazing and actual grazing treatments (Klein et al. 2004; 2007), and showed the urine/dung deposition and tread by animals' effect on soil and plant is limited. We believe that removal of litter can stimulate the effect of winter grazing, which has been demonstrated by previous studies (Zhu et al. 2015; Che et al. 2018). We had improved the description of the winter grazing treatment and make it more clearly, please see lines 159-174.

[Comments] Lines 195-196: Please revise this title.

[Responses] Done, please see lines 204-205.

[Comments] Line 201 and line 235: The monthly mean temperature was 9.7 C in August, but the slurry was incubated under 28 C. The incubation temperature is nearly two times greater than the mean temperature. How would this artificial effect modulate the responses of the measured indices?

[Responses] The incubation experiment was measured the soil ability/potential of N2O production, not the field N2O flux. The method was provided a good condition for the soil microbial, eg. relative high incubation temperature, and added some substrate, so that can inspire the activities of nitrifying and denitrifying microorganism and show the ability N2O produce by nitrifying and denitrifying microorganism in soils. The method and the incubation condition was commonly used to measure the nitrification and denitrification enzyme activity and proved to be useful (Smith and Tiedje, 1979; Simek and Hopkins, 1999; Chroňáková et al. 2009; Cantarel et al. 2012).

[Comments] Line 203: What "them" stands for?

[Responses] "them" stands for slurry, we clarified it. Please see lines 212.

[Comments] Line 220: nitrification again?

[Responses] It is denitrification, we corrected it. Please see lines 230.

[Comments] 3. Results and Discussion Lines 286-291: I can't see how you can jump from nitrification or denitrification potentials to assessing the contributions of bacterial and fungi to potential N2O emissions. Nitrification or denitrification potentials should

not be regarded as N2O productions especially emissions by nitrification or denitrification. From this sense, the discussion section should be rewritten thoroughly.

[Responses] Most studies mainly focused on the contribution of bacterial nitrification and denitrification to potential N2O emissions. Because numerous studies have shown that fungal nitrification and denitrification can play an important role in N2O production. Therefore, in this study we aimed to quantify the contribution of fungal and bacterial to potentials of N2O from nitrification and denitrification. Because the contribution of fungal nitrification and denitrification was higher than bacteria's (Fig. 5) in control treatment, this indicates that the fungi played the major role in potential N2O emissions.

We agreed with the referee that the nitrification or denitrification potentials should not be regarded as N2O productions especially emissions. In this study, we mainly focused on the mechanism of N2O produce process and distinguished the role of bacteria and fungi in N2O produce process. In the new version, we rewrote the discussion section and related sections to avoid the misunderstanding.

**Supplement:**

[revised manuscript text omitted]

Fig. 2

[Figure]

Fig. 3

[Figure]

Fig.5

[Figure]

---

## Author Comment (AC5) · 30 Apr 2018

Thank you for you suggestions. We have revised our manuscript "Fungi regulate response of N2O production to warming and grazing in a Tibetan grassland", based on your comments. We have carefully addressed each comment and our responses to these comments are listed the below. The attachments are the manuscript which had improved as your suggestions. We hope that all necessary revisions have been made. However, we would be prepared to make further revisions and modifications if required.

Responses to the Reviewer's comments:

[Comments] (1) The statistical analysis and reporting are weak. Is there any real field replication, excluding any pseudo replication? What was the power of the statistical

test? Statistical differences among different treatments were not reported for all the sub-plots. Additionally, along with p values, standard Error of the mean difference may need to be reported in the plots to understand the differences between the treatment means better.

[Responses] Yes, we had real field replication, our site is a two-way factorial design (warming and grazing) was used with four replicates of each of four treatments. In total, 16 plots of 3-m diameter were fully randomized throughout the study site. We had shown it in our manuscript, please see the lines 144-148.

About the statistical differences among different treatments, it was also mentioned by other reviewer, as his suggestion, we removed the different letters from the Figs. 1b and 4e to avoid the misunderstandings. We also showed the two-way ANOVA results in Table 1 to give more details of statistical analysis in our manuscript. Please see lines 581-643.

[Comments] (2) It was not clear how were the relative contributions of bacteria and fungi in nitrification, denitrification and total N2O production derived from the total respective measurements? The methods need to be clear and reproducible.

[Response] For the contribution of bacteria and fungi to total nitrification enzyme activity was calculated it by the ratio of BNEA or FNEA to BNEA+FNEA; the contribution of bacteria and fungi to total potential of N2O production from denitrification was calculated it by the ratio of BDEA or FDEA to BDEA+FDEA. In the new version, we added the description in Materials and Methods". Please see the lines 255-258.

[Comments] (3) In addition to the present results of the relative contribution of bacteria and fungi in nitrification and denitrification, the definite mechanisms for bacterial and fungal pathways of nitrification and denitrification need to present to demonstrate the change in the pathway of N2O production under the warming treatment. A definite mechanism of shifting in the relative contribution of bacteria and fungi in N2O production would help the reader to understand the present results in a systematic way, particularly under the warming treatment. This would also help to explain and understand the underline reasons of changing the pathway of N2O production between bacteria and fungi under warming.

[Responses] It is the two reasons that lead to the changes of fungal and bacterial pathways for N2O emissions by warming. Firstly, the increased of soil temperature directly reduce fungal activity but increase bacterial activity, because fungi prefer the cold environment compared with bacteria. Secondly, warming indirectly reduce fungal activity but increase bacterial activity through increased soil inorganic N and decreased soil organic N in our site, please see lines 358-363, because fungi prefer higher organic N environment while bacteria prefer higher inorganic N environment. All these changes caused the fungal and bacterial pathways for N2O emissions changed in different directions under warming. We have improved the manuscript and make sure the underlying mechanisms is clearly, please see lines 352-365.

[Comments] (4) It was also not clear why the effects of warming on relative contribution of bacteria and fungi on nitrification, denitrification were diluted when warming treatment was combined with grazing, for example in fig 5?

[Responses] Yes, the effects of warming on relative contribution of bacteria and fungi on nitrification, denitrification were diluted when warming treatment was combined with grazing in our results. We had discussed in above that warming changed the pathway of N2O production potential mainly through alter the soil temperature and the soil inorganic and organic N content. In our results, (WG) also reduced the positive effect of (W) on the soil temperature (Fig. 1b), and showed the trend of reduced the negative effect of (W) on the TC, TN and NO3- content although the statistical analysis were not significantly (Fig. 2), moreover, the soil dissolved organic nitrogen content was significantly diluted when warming treatment was combined with grazing (data not shown), so the effect of (WG) on soil temperature and the substrate concentration caused the effects of warming on relative contribution of bacteria and fungi on nitrification, denitrification were diluted when warming treatment was combined with grazing.

Please also note the supplement to this comment:
https://www.biogeosciences-discuss.net/bg-2017-552/bg-2017-552-AC5-
supplement.pdf

**Supplement:**

[revised manuscript text omitted]

Fig. 2

[Figure]

Fig. 3

[Figure]

Fig.4

[Figure]

Fig.5

[Figure]

---

## Author Comment (AC6) · 30 Apr 2018

Thank you for you suggestions. We have revised our manuscript "Fungi regulate response of N2O production to warming and grazing in a Tibetan grassland", based on your comments. We have carefully addressed each comment and our responses to these comments are listed the below. The attachments are the manuscript which had improved as your suggestions. We hope that all necessary revisions have been made. However, we would be prepared to make further revisions and modifications if required.

Responses to the Reviewer's comments:

[Comments] 1. The description of experimental desigh is not clear, particularly, there is a confusing in introducing winter grazing treatment. What is the reason for the selection

of winter grazing treatment in present study? Tibetan grassland is experienced to be covered by snow, frozen soils, and the grass should be withered in winter. In the same plots, the ecological effects of winter grazing should be interferenced by previous different grazing treatments (lines 153-156). How to avoid it?

[Responses] Sorry, our previous description caused the misunderstanding by the referee. In the new version, we clarified why we used winter grazing. On the Qinghai-Tibet plateau, winter grazing is very commonly and alpine meadows are generally classified into two grazing seasons, i.e. warm season grazing from June to September and cold season grazing from October to May even the grassland was covered by snow (Cui et al., 2015). Winter pasture contributed about 40

We had improved the description of the winter grazing treatment and make it more clearly, please see lines 159-174.

[Comments] 2. Potential total nitrification/denitrification for N2O emission rate from incubation experiment is not a "real" rate of N2O emission under the field conditions. In terrestrial ecosystems, soil temperature, moisture, pH, soil N availability, and DOC etc. are generally considered as the major factors of controlling N2O emissions. For this study, the lack of field simultaneous monitoring data of N2O rates is a critical issue. Although the authors tried to cite the previous results for discussion, the conclusion obtained from an incubation experiment is still not general acceptable.

[Responses] We fully agree with the referee that the fungal and bacterial enzyme activities cannot be shown as the result of N2O emissions. The measurements under laboratory incubation reflected the potential ability of the soil fungal and bacterial activities in nitrification and denitrification because such laboratory incubation could avoid the impacts of various confounding factors and well clarify the mechanism responsible for N2O produce process. For the lack of field simultaneous monitoring data of N2O rates, because our study was focused on the microbial mechanism responsible for N2O produce process but not for the N2O flux, so we think the field N2O emission is not necessary. There are also a series of studies showed the microbial mechanism responsible for N2O produce process and conclusions by incubation experiment, eg. Zhong et al. (2015, 2017); Huang et al. (2017); Marusenko et al. (2013); Attard et al. (2011) and so on. At revised version, we clarified that our measurements in the laboratory indicated the potential emission to reveal the mechanism responsible for N2O produce process but not the field emission.

[Comments] 3. The underlying mechanisms that fungal and bacterial pathways for controlling N2O emissions remain unknown. The authors need to elaborate the relative contributions of fungi and bacteria in nitrification and denitrification processes of N2O productions.

[Responses] It is the two reasons that lead to the changes of fungal and bacterial pathways for N2O emissions by warming. Firstly, the increased of soil temperature directly reduce fungal activity but increase bacterial activity, because fungi prefer the cold environment compared with bacteria. Secondly, warming indirectly reduce fungal activity but increase bacterial activity through increased soil inorganic N and decreased soil organic N in our site, please see lines 350-355, because fungi prefer higher organic N environment while bacteria prefer higher inorganic N environment. All these changes caused the contribution of fungi in nitrification and denitrification was reduced by warming, but the contribution of bacteria in nitrification and denitrification was increased by warming (Fig.5), then due to the fungal and bacterial pathways for N2O emissions was changed in different directions under warming. We have improved the manuscript and make sure the underlying mechanisms is clearly, please see Lines 353-366.

[Comments] 4. Line 130-131: The symbol oC is not correct.

[Responses] Thank you for your suggestion. We had corrected itïijŇ please see lines 136-137.

[Comments] 5. There are several mistakes in English writing, which should be revised throughout the text.

[Responses] In the new version, we almost rewrote the manuscript and asked a native English speaker Miss Ri Weal to polish the language errors. We hope the new version is easy to read and follow.

**Supplement:**

[revised manuscript text omitted]

Fig. 2

[Figure]

Fig. 3

[Figure]

Fig.4

[Figure]

Fig.5

[Figure]

---

## Author Response (AR1)

June 17th, 2018

Dear Editor,

We have revised our manuscript "Fungi regulate response of $N_2O$ production to warming and grazing in a Tibetan grassland", based on the comments of all the reviewers. We have carefully addressed each comment and our responses to these comments are listed in the below. We hope that all necessary revisions have been made. However, we would be prepared to make further revisions and modifications if required.

Yours sincerely,

Dr. Wenchao Ma
43-b-301, Department of Environmental
Engineering School of Environmental
Science and Technology
Tianjin University
Yaguan Road 135#,
Haihe Education Park,
Jinnan District, Tianjin 300350 China

**Responses to the Reviewer's comments:**

**To Prof. Feng**

[Comment]- This is a concise and nicely written paper, focusing on fungal and bacterial contributions to potential $N_2O$ emissions in an alpine grassland in response to warming and grazing treatments in the field. The authors report several interesting observations, including an increased bacterial enzyme activity and a decreased fungal enzyme activity for $N_2O$ emissions under warming. The results have immediate implications for GHG emissions under the scenario of climate change. I have several suggestions for the authors to consider in order to improve the manuscript.

1. Although the authors showed that fungal and bacterial pathways for $N_2O$ emissions changed in different directions under warming, the underlying mechanisms, or causes remain unknown. In Line 321-322, it is mentioned that increased $NO_3^-$-N may inhibit fungal growth. Can you elaborate more? Also, did warming affect soil moisture contents or dynamics compared to the control? If so, how would moisture change affect fungal versus bacterial communities? In the end, I am interested in the driving force leading to the observed changes, it is direct warming effect or indirect effect mediated by other factors? Unless we know answers to these questions, we can hardly speculate on the future changes.

**[Responses]- We thank the reviewer for the kind suggestion.**

**For "fungal and bacterial pathways for $N_2O$ emissions changed in different directions under warming, the underlying mechanisms or causes remain unknown."**

   **It is the two reasons that lead to the changes of fungal and bacterial pathways for $N_2O$ emissions by warming. Firstly, the increased of soil temperature directly reduce fungal activity but increase bacterial activity, because fungi prefer the cold environment compared with bacteria. Secondly, warming indirectly reduce fungal activity but increase bacterial activity through increased soil inorganic N and decreased soil organic N in our site, please see Lines 352-365, because fungi prefer higher organic C/N environment while bacteria prefer higher inorganic C/N environment. All these changes caused the fungal and bacterial pathways for $N_2O$ emissions changed in different directions under warming.**

**We have improved the manuscript and make sure the underlying mechanisms is clearly, please see Lines 352-377.**

**For "In Line 321-322, it is mentioned that increased $NO_3^-$-N may inhibit fungal growth. Can you elaborate more?".**

We showed more data to support our findings, at our site, not only the soil inorganic N was increased, as reflected by soil $NO_3^-$-N concentration (Fig. 2a and 2b); but also the soil dissolved organic nitrogen was significantly decreased from 48 to 41 mg kg$^{-1}$ (P<0.04). Moreover the soil labile C and N was also found significantly decreased by warming (Rui et al., 2012). Warming indirectly reduce fungal activity but increase bacterial activity through increased soil inorganic N and decreased soil organic N in our site, please see Lines 358-365.

For "did warming affect soil moisture contents or dynamics compared to the control? If so, how would moisture change affect fungal versus bacterial communities?".

Yes, warming significantly decreased soil moisture at our site (Fig. 1), but we do not think warming affected fungal versus bacterial communities through the soil moisture. Although the fungi prefer the relative dry soil condition, the NEA and DEA from fungi were not increased, while the NEA and DEA from bacteria were not increased in the warming treatment. This might be duo to the fact that warming induced changes in in soil moisture is not great enough to affect the fungal and bacterial community.

For "I am interested in the driving force leading to the observed changes, it is direct warming effect or indirect effect mediated by other factors?"

We believe that warming directly affected the fungal versus bacterial communities due to the increase of the temperature. Additionally, warming also indirectly mediated the fungal versus bacterial communities through the changes in the substrate. We had dicussed it in the first section, improved the manuscript and make sure the underlying mechanisms is clearly, please see Lines 352-377.

[Comment]- 2. Speaking of future predictions, I think it should be emphasized that measurements made here were potential rather than "real" emissions in the field. A critical requirement for denitrification to occur is anoxic or sub-oxic conditions. Therefore, I would think that $N_2O$ emissions more depend on the hydrological or redox conditions of the soil. Observations of fungal and bacterial enzyme activity changes in the lab may or may not apply to the field observations, depending on how warming affects soil moisture.

[Responses]-For "Speaking of future predictions, I think it should be emphasized that measurements made here were potential rather than "real" emissions in the field."

We fully agree with the referee that the fungal and bacterial enzyme activities cannot be shown as the result of $N_2O$ emissions. The measurements under laboratory incubation reflected the potential ability of the soil fungal and bacterial activities in nitrification and denitrification because such laboratory incubation could avoid the impacts of various confounding factors and well clarify the mechanism responsible for $N_2O$ emission. At revised version, we clarified that our measurements in the laboratory indicated the potential emission.

For "A critical requirement for denitrification to occur is anoxic or sub-oxic conditions. Therefore, I would think that $N_2O$ emissions more depend on the hydrological or redox conditions of the soil."

  Yes, we also fully agree with the referee that anoxic or sub-oxic conditions and soil moisture is very important for $N_2O$ emissions. For hydrological or redox conditions, because we did not measure it, so it is hard to dicussed it directly, but it was mainly influenced by soil moisture, the soil moisture was showed in Fig. 1c and Table 1.

For "Observations of fungal and bacterial enzyme activity changes in the lab may or may not apply to the field observations, depending on how warming affects soil moisture."

  The observations of fungal and bacterial enzyme activities were also not applied as the field emissions, they were used to clarify the mechanism responsible for $N_2O$ emission. In our stie, the filed $N_2O$ emission in 2011-2012 was shown in the manuscript. And the laboratory measurements of the total nitrification and denitrification enzyme activities all were the same with the filed $N_2O$ emission at our site (Zhu et al. 2015; Fig. 4c and 4f; Table 1), which showed it could well clarify the mechanism responsible for $N_2O$ emission.

For "depending on how warming affects soil moisture".

  Although warming significantly decreased the soil moisture at our site, the field $N_2O$ emission, total nitrification and denitrification enzyme activity did not change as a result of warming (Zhu et al. 2015; Fig. 4c and 4f; Table 1). It might be due to the fact that the changes in soil moisture by warming was not great enough to lead to a detectable difference in field $N_2O$ emission, total nitrification and denitrification enzyme activity.

[Comment]-Some minor points: Line 163: I notice that there was no field replicate for the measurement?

[Responses]- In this study, we used in field replicates. There were four replicates for each of four treatments. Therefore, we had 16 plots in total. We collected soil samples from each plot. We made detailed description on how to collect soil in the revised version, please see lines 144-148.

[Comment]-Line 223: N2 not N.

[Responses]- Corrected, please see lines 243.

[Comment]-Line 227: Why only three time points for the denitrification measurement versus 5 points for nitrification?

[Responses]-For DEA incubation experiment, we collected at least 12 ml gas for $N_2O$ concentration measuring. If too many times were used to collect $N_2O$, it would change the incubation pressure and influence the responsibility of the experiment. So, we only collected 3 times in the incubation experiment. But for NEA incubation experiment, it does not matter. Additionally, different sampling times for NEA and DEA should have little effect on the reliability of our results because this study did not aim to distinguish the contribution of total nitrification and denitrification to $N_2O$ emissions. Here we just estimated nitrification enzyme activity by analyzing the change of $NO_2^-+NO_3^-$ concentration after incubation, see lines 222-224 and denitrification enzyme activity by analyzing the change of $N_2O$ concentration after incubation, see lines 249-250. Overall. we only compared NEA and DEA among all treatments, respectively.

[Comment]-Lines 285 and 292: NEA, DEA, FDEA, BDEA: : :not used in the previous text.

[Responses]-For NEA, DEA, we changed it to TNEA and TDEA. They were used in the previous version, please see lines 206 and 229-230.

[Comment]-Line 304: I don't think IC is much higher in Haibei soils than some temperate grassland soils in Mongolia. IC contents are dependent on soil pHs…
[Responses]-Thank you for your suggestion. We corrected it in the new version, please see lines 325-329.

**To Anonymous Referee #3**

[Comment]- This study reports the effect of warming and grazing on soil biotic contribution to $N_2O$ production in a Tibetan grassland, by examining a long-term (over 10 years) experiment combined with an incubation experiment. Their results indicated that fungi could be the main source for $N_2O$ production potential in the Tibetan alpine grasslands. Overall, the manuscript is of interest and generally well written. But there are some concerns and unclear points that should be addressed prior to publication.
Please find some more detailed comments below.

[Comment] Lines 162-164, is it enough to collect only 5 cores for each soils?

**[Responses] For soil sampling, "randomly collected" was used to reduce the spatial heterogeneity. At our site, soil samples were collected using this method in all related experiments because the plot area used for warming was limited. However, this sampling method was proved to be suitable which can be found in a series of our published papers in Ecology (Wang et al. 2012), Global change biology (Luo et al. 2010), Journal of soils and sediments (Rui et al. 2012) and so on.**

[Comment] How the authors draw the contribution of bacteria and fungi to total nitrification enzyme activity and total denitrification enzyme activity as shown in Fig. 5? I cannot find the specific description in the section "Materials and Methods".

**[Responses] Thank you for your comment. In the new version, we added the description in Materials and Methods". Please see the lines 256-259.**

[Comment] Line 257, "Fig. 1A" should be changed to "Fig. 1a", based on the Figure 1. Also, the authors should revised it throughout the main text.

**[Responses] Corrected.**

[Comment] Line 259, "soil moisture" should be changed to "The average soil moisture".

**[Responses] Corrected. Please see the lines 272.**

[Comment] In Figs. 1 and 4, why significant differences were only shown in Figs. 1b and 4e rather than all of subfigures?

**[Responses] We have removed different letters from the Figs. 1b and 4e. The two-way ANOVA results in all figures were enough. Please see lines 591-646.**

**To Anonymous Referee #1**

Comments on Zhong et al. for Biogeosciences Discussion

[Comment] This manuscript presents an interesting study on the response of an alpine grassland ecosystem to warming and grazing in the period of 10 years. $N_2O$ production via variable microbial components was the main focus. It is written concisely and easy to understand. However, regarding the experiment design and interpretation of the dataset, I believe that there is still more to improve before it could be published. Despite their investigation into multiple treatments and parameters, the authors need to provide more field evidence and literature comparison to reach a convincing conclusion. Throughout the whole manuscript, the authors seem to mix up denitrification enzymatic activity and $N_2O$ production. If the inhibitors applied in the experiments to determine denitrification rates also inhibit $N_2O$ reduction to $N_2$, the $N_2O$ production should rather represent potential denitrification rates. If $N_2O$ reduction was not inhibited during the experiment, the results could not be noted as "denitrification rates". Please clarify this key point and make revision accordingly. The methods determining these rates should be described in more details in M&M.

[Responses] Based on the reviewer's suggestions, we provided more field data and literature to support our conclusion. The field $N_2O$ emission in 2011-2012 at our site (Zhu et al. 2015) was referenced in our manuscript, please see lines 347-349 and lines 393-396. We also added the mean temperature and rainfall data during the sampling year and months;the soil dissolved organic nitrogen data in our manuscript, please see lines 136-138 and lines 360-363. Because these data were obtained by other colleagues, we cannot present them as figures in the current study. The filed $N_2O$ emission supported our conclusion of warming had no effect on total nitrification and potential of $N_2O$ production from denitrification. The soil dissolved organic nitrogen data supported our conclusion of warming reduced the potential of $N_2O$ from fungi because of the reduction of organic substrates.

We also showed more references to supports our conclusions, e.g. Zhu et al. (2015) to support our conclusion of warming had no effect on total nitrification and potential of $N_2O$ production from denitrification; the results of Zhu et al. (2015), Krümmelbein et al. (2009) and Steffens et al. (2008) supported our conclusion of winter grazing had little effect on environment because the soil is frozen in winter and often covered with snow and grazing has little effect on soil conditions, please see lines 347-349 and 389-391.

To determine potential denitrification rates, we incubated soil samples under anaerobic condition and did not add any inhibitor to inhibit $N_2O$ reduction to $N_2$ process. Therefore, our results only can be presented as the potential of $N_2O$ emission from denitrification. We have clarified this in M&M, please see lines 204-259.

[Comment] Line 111: "To clarify whether fungi control the $N_2O$ production process" is misleading as Fungi contributes anyway; I assume that the authors wish to clarify the "role of fungi in $N_2O$ production process"

**[Responses] Done as your suggestion. please see lines 117.**

[Comment] Line 161–162: Please explain this; why do you see the effects on ecosystem level despite that plot size are 3 m? Any data to support this?

**[Responses] This is really good question. The plot size used for warming treatments are generally small, less than 1 $m^2$ (Cantarel et al. 2012) to more than 10 $m^2$ (Long et al. 2015). These studies well showed the effects of treatments on ecosystem (Cantarel et al. 2012; Long et al. 2015). In this study, the size of our plots was considered according to three points: 1) A little big size was used because grazing was involved. Although the size of plot might affect the animal feeding activities, all experimental sheep were fenced into three additional 5\*5 m fenced plots for one day before the beginning of the grazing experiment to help them adapt to small plots for reducing the experimental error. 2) The warming efficiency and cost (we used the infrared heaters in warming treatments for increasing soil temperature) was another factor; and 3) the species composition and vegetation coverage is even in this grassland. Previous publications (Wang et al. 2012 Ecology, Luo et al. 2010 Global Change Biology, Luo et al. 2009 Soil Biology and Biochemistry, Rui et al. 2012 Journal of Soils and Sediments) from this study have demonstrated that the plot size can show the effects on ecosystem level.**

[Comment] Line 165: If 10 years' warming and grazing treatment was done, why was only one sampling of soils by the end of 10 years' treatment? Have you considered the soil heterogeneity between control and treatment plots since the beginning of treatments?

**[Responses] Only one sampling of soils was done by the end of 10 years treatment. The reason is that this is the first time for us to pay attention to the contribution of fungi and bacteria to $N_2O$ production based on recent research advances and fresh soil is required for microbial analysis especially for the incubation experiment. A thorough understanding about the long-term impact of warming and grazing on soil fungal nitrification and denitrification from alpine meadow grassland requires further investigation through multi-sampling during a long period. We mentioned this limitation in Discussion, please see lines 401-404. Additionally, we considered the soil heterogeneity between control and treatment plots since the beginning of treatments. There is no difference between treatments the beginning of this experiment. To reduce the soil heterogeneity, all the plots were assigned in a complete randomized block.**

**For "soil heterogeneity between control and treatment plots since the beginning of treatments?". We think the spatial heterogeneity was exit in everywhere.**

[Comment] Line 166: Including or excluding organic layer? Please specify.

**[Responses] Done as your suggestion. please see lines 176.**

[Comment] Line 225 – 226: 100% of water-holding capacity could favor denitrification; however, it may not likely represent field condition, which is usually drier. Please justify your choice of such incubation condition.

**[Responses] The incubation experiment was used to show the potential of $N_2O$ produce from denitrification of soil, it cannot be represented as the $N_2O$ production of field. The 100% of water-holding capacity was provided an relative good environment for denitrification so that can inspire the activities of denitrifying microorganism and show the ability $N_2O$ produce by denitrifying microorganism in soils. The method and the incubation condition was commonly used to measure the denitrification enzyme activity and proved to be useful (Smith and Tiedje, 1979; Simek and Hopkins, 1999; Chroňáková et al. 2009; Cantarel et al. 2012).**

[Comment] Line 294 – 298: Use present tense: use "is" to replace "was".

**[Responses] Done as your suggestion. Please see lines 307-309.**

[Comment] Line 298: Change "who" to "whom".

**[Responses] Done as your suggestion. Please see lines 312.**

[Comment] Line 314 to 315: When comparing the studied alpine grassland to temperate grassland, how do come to the conclusion that the lower inorganic C and N contents in soil were due to larger fungal contribution to $N_2O$ production? What about the higher mineralization rates in the temperate systems? In addition, the control of inorganic C or N levels in soil could be also related to biomass uptake and turnover. Please clarify it and avoid such speculation.

**[Responses] We fully agree with the referee that the lower inorganic C and N contents in soil based on observations from alpine grassland to temperate grassland cannot come to the conclusion. In the new version, we removed the sentence and improved this part to avoid such speculation. Please see lines 325-329.**

[Comment] Line 324: "common" and "globally" do not fit together; please revise.

**[Responses] Done as your suggestion. Please see lines 334.**

[Comment] Line 348–349: "gene abundance of fungi was not changed" against treatments; how do you reconcile your finding with the hypothesis?

[Responses] The gene abundance of fungi was not changed by warming, but warming changed FNEA and FDEA. Such inconsistency between gene abundance of fungi and FNEA/FDEA might be explained by the fungal gene abundance not providing information on real-time process rates. The reason is that process rates are largely dependent on environmental conditions. Fluctuations in environmental conditions can cause rapid changes in real-time process rates, but do not necessarily affect gene abundance (Zhong et al. 2014). We have improved it in the new version, please see lines 368-374.

**To Anonymous Referee #5**

I have some major concerns as shown below:

[Comment] The experimental design is not acceptable. Firstly, why did you choose "winter grazing"? There seems no explanation. The temperature should be too low to let the animal grazing out of the field in winter. Additionally, the grassland is expected to be covered by snow and the grasses should be withered in winter. Secondly, the description of the treatment is really confusing. Winter grazing should be used in the current study, but "For grazing treatments, the grazing treatments in this site were used for summer grazing treatments until 2010, from 2011 to 2015, there was no grazing during the summer, and grazing was replaced by cutting and removing about 50% of the litter biomass in October and the following March each year to simulate winter grazing" (lines 153-156). To be honest, I can't understand the experimental design at all. In addition, grazing can't be simulated by cutting or mowing, since grazing involves tread and urine/dung deposition. Even the land is very hard due to freezing in winter, tread by animals would result in different effects on the plant communities.

**[Responses] Sorry, our previous description caused the misunderstanding by the referee. In the new version, we clarified why we used winter grazing. On the Qinghai-Tibet plateau, winter grazing is very commonly and alpine meadows are generally classified into two grazing seasons, i.e. warm season grazing from June to September and cold season grazing from October to May even the grassland was covered by snow (Cui et al., 2015). Winter pasture contributed about 40% of the grazed area in Qinghai–Tibet Plateau (Fan et al. 2010). See lines "66-174".**

**In the new version, we clarified our design. During 2006-2010 summer grazing treatments was used to explore the effects of warming and grazing on ecosystem during the growing seasons (Luo et al. 2010; Hu et al. 2010; Wang et al. 2012). Considering strong disturbance, grazing was stopped during 2011-2015. Given the importance of winter grazing, winter grazing during the non-growing seasons was further investigated (Zhu et al. 2015; Che et al. 2018). We agree with the referee that grazing cannot be simulated by cutting or mowing since grazing involves tread and urine/dung deposition. However, during winter, such effects could be very small because soil and dung are frozen and tread has little effect on soil. Actually, we had examined how clipping simulated the effects of actual grazing before we established four replicated "actual grazing treatments" compare with the "simulated grazing treatments", the soil and plant all showed no difference between simulated grazing and actual grazing treatments (Klein et al. 2004; 2007), and showed the urine/dung deposition and tread by animals' effect on soil and plant is limited. We believe that removal of litter can stimulate the effect of winter grazing, which has been demonstrated by previous studies (Zhu et al. 2015; Che et al. 2018). We had improved the description of the winter grazing treatment and make it more clearly, please see lines 159-174.**

[Comment] I can't see how you can jump from nitrification or denitrification potentials to assessing the contributions of bacterial and fungi to potential $N_2O$ emissions. Nitrification or denitrification potentials should not be regarded as $N_2O$ productions especially emissions by nitrification or denitrification. From this sense, the discussion section should be rewritten thoroughly.

**[Responses] Most studies mainly focused on the contribution of bacterial nitrification and denitrification to potential $N_2O$ emissions. Because numerous studies have shown that fungal nitrification and denitrification can play an important role in $N_2O$ production. Therefore, in this study we aimed to quantify the contribution of fungal and bacterial to potentials of $N_2O$ from nitrification and denitrification. Because the contribution of fungal nitrification and denitrification was higher than bacteria's (Fig. 5) in control treatment, this indicates that the fungi played the major role in potential $N_2O$ emissions.**

**We agreed with the referee that the nitrification or denitrification potentials should not be regarded as $N_2O$ productions especially emissions. In this study, we mainly focused on the mechanism of $N_2O$ produce process and distinguished the role of bacteria and fungi in $N_2O$ produce process. In the new version, we rewrote the discussion section and related sections to avoid the misunderstanding.**

[Comment] The manuscript is not well prepared. There are lots of writing issues throughout the manuscript. I only presented few of them since there are too many.

**[Responses] In the new version, we almost rewrote the manuscript and asked a native English speaker Miss Ri Weal to polish the language errors. We hope the new version is easy to read and follow.**

[Comment] Abstract Lines 44-46: The treatments should be described briefly in the abstract to increase the readability. Additionally, some key information about the method should be presented.

**[Responses] Done. Please see lines 44-48.**

[Comment] Lines 46-52: The values should be presented with uncertainties, e.g., standard error, standard deviation or 95% confidence interval. Similarly, the relevant values in the text should be presented with uncertainties.

**[Responses] Done. Please see lines 48-49, 54-55, and Fig.5.**

[Comment] Lines 46-47: Were these values got from the control?

**[Responses] Yes, these values are obtained from the control. We clarified this in the new version, please see lines 49.**

[Comment] Lines 49-52: Suggest rephrase these sentences in such way: "However, warming significantly increased the enzyme activity of bacterial nitrification and denitrification to 53% and 55%, respectively, but decreased enzyme activity of fungal nitrification and denitrification to 47% and 45%, respectively. Winter grazing had no such effects."

**[Responses] Done. Please see lines 53-55.**

[Comment] Lines 52-54: How could you make this conclusion? Under what conditions do soil fungi contribute more to N2O production? This sentence is of course not clear. If the conclusion is obtained based on results from the control, it should be put somewhere after lines 46-47. Additionally, can you make such a strong conclusion based on an incubation experiment?

**[Responses] Thank the referee for pointing out the question. We rewrote the abstract as the referee suggested, please see lines 40-62. Our conclusion was based on the role of fungi and bacteria in $N_2O$ produce process by the incubation experiment but not in $N_2O$ emissions. In the new version, we clarified this, please see lines 1-62.**

[Comment] Lines 56-58: This should not be put in the abstract as a key implication since it should be regarded as a fact.

**[Responses] Done. Please see lines 58-60.**

[Comment] Line 59-60: This sentence should be rephrased since some grammar issue exists. For example, "lead to refine: : :." is not correct. Overall, the abstract needs substantial revision.

**[Responses] Done. Please see lines 40-62.**

[Comment] Introduction Line 66: not clear what does "it" refer to.

**[Responses] "it" refer to $N_2O$ emission, we clarified it, please see lines 67-69.**

[Comment] Lines 67-69: This sentence needs substantial revision.

**[Responses] Done, please see lines 69-71.**

[Comment] Line 122: Why did you choose "winter grazing"? There seems no explanation. The temperature should be too low to let the animal grazing out of the field in winter.

Additionally, the grassland is expected to be covered by snow and the grasses should be withered in winter.

**[Responses] Sorry, our previous description caused the misunderstanding by the referee. In the new version, we clarified why we used winter grazing. On the Qinghai-Tibet plateau, winter grazing is very commonly and alpine meadows are generally classified into two grazing seasons, i.e. warm season grazing from June to September and cold season grazing from October to May even the grassland was covered by snow (Cui et al., 2015). Winter pasture contributed about 40% of the grazed area in Qinghai–Tibet Plateau (Fan et al. 2010), please see lines 66-174.**

[Comment] M & M Lines 130-131: The symbol C is not correctly used.

**[Responses] Done, please see lines 136-137.**

[Comment] Lines 131-132: over 80% of which?

**[Responses] Over 80% of total rainfall, we clarified it, please see lines 138.**

[Comment] Lines 133-134: Please clearly present the soil classification systems and the references.

**[Responses] Done, please see lines 139.**

[Comment] Lines 134: There should be a space between the word and the parentheses here and in other sentences or Figures (Please check the figures as well).

**[Responses] Done, please see lines 142 and the caption of figures.**

[Comment] Line 139: The indent here is not consistent with other paragraphs. Please keep consistency.

**[Responses] Done, please see lines 144.**

[Comment] Line 146: delete was.

**[Responses] Done, please see lines 151.**

[Comment] Lines 153-156: The description is really confusing. According to the above paragraph, winter grazing was used in the current study, but "For grazing treatments, the grazing treatments in this site were used for summer grazing treatments until 2010, from 2011 to 2015, there was no grazing during the summer, and grazing was replaced by cutting and removing about 50% of the litter biomass in October and the following March each year to simulate winter grazing". To be honest, I can't understand the experimental design at all. In addition, grazing can't be simulated by cutting or mowing, since grazing involves tread and urine/dung deposition.

**[Responses] In the new version, we clarified our design. During 2006-2010 summer grazing treatments was used to explore the effects of warming and grazing on ecosystem during the growing seasons (Luo et al. 2010; Hu et al. 2010; Wang et al. 2012). Considering strong disturbance, grazing was stopped during 2011-2015. Given the importance of winter grazing, winter grazing during the non-growing seasons was further investigated (Zhu et al. 2015; Che et al. 2018). We agree with the referee that grazing cannot be simulated by cutting or mowing since grazing involves tread and urine/dung deposition. However, during winter, such effects could be very small because soil and dung are frozen and tread has little effect on soil. Actually, we had examined how clipping simulated the effects of actual grazing before we established four replicated "actual grazing treatments" compare with the "simulated grazing treatments", the soil and plant all showed no difference between simulated grazing and actual grazing treatments (Klein et al. 2004; 2007), and showed the urine/dung deposition and tread by animals' effect on soil and plant is limited. We believe that removal of litter can stimulate the effect of winter grazing, which has been demonstrated by previous studies (Zhu et al. 2015; Che et al. 2018). We had improved the description of the winter grazing treatment and make it more clearly, please see lines 159-174.**

[Comment] Lines 195-196: Please revise this title.

**[Responses] Done, please see lines 204-205.**

[Comment] Line 201 and line 235: The monthly mean temperature was 9.7 C in August, but the slurry was incubated under 28 C. The incubation temperature is nearly two times greater than the mean temperature. How would this artificial effect modulate the responses of the measured indices?

**[Responses] The incubation experiment was measured the soil ability/potential of $N_2O$ production, not the field $N_2O$ flux. The method was provided a good condition for the soil microbial, eg. relative high incubation temperature, and added some substrate, so that can inspire the activities of nitrifying and denitrifying microorganism and show the ability $N_2O$ produce by nitrifying and denitrifying microorganism in soils. The method and the incubation condition was commonly used to measure the nitrification and denitrification enzyme activity and proved to be useful (Smith and Tiedje, 1979; Simek and Hopkins, 1999; Chroňáková et al. 2009; Cantarel et al. 2012).**

[Comment] Line 203: What "them" stands for?

**[Responses] "them" stands for slurry, we clarified it. Please see lines 212.**

[Comment] Line 220: nitrification again?

**[Responses] It is denitrification, we corrected it. Please see lines 230.**

[Comment] Results and Discussion
Lines 286-291: I can't see how you can jump from nitrification or denitrification potentials to assessing the contributions of bacterial and fungi to potential N2O emissions. Nitrification or denitrification potentials should not be regarded as N2O productions especially emissions by nitrification or denitrification. From this sense, the discussion section should be rewritten thoroughly.

**[Responses] Most studies mainly focused on the contribution of bacterial nitrification and denitrification to potential N$_2$O emissions. Because numerous studies have shown that fungal nitrification and denitrification can play an important role in N$_2$O production. Therefore, in this study we aimed to quantify the contribution of fungal and bacterial to potentials of N$_2$O from nitrification and denitrification. Because the contribution of fungal nitrification and denitrification was higher than bacteria's (Fig. 5) in control treatment, this indicates that the fungi played the major role in potential N$_2$O emissions.**

**We agreed with the referee that the nitrification or denitrification potentials should not be regarded as N$_2$O productions especially emissions. In this study, we mainly focused on the mechanism of N$_2$O produce process and distinguished the role of bacteria and fungi in N$_2$O produce process. In the new version, we rewrote the discussion section and related sections to avoid the misunderstanding.**

The present manuscript, entitled "Fungi regulate response of $N_2O$ production to warming and grazing in a Tibetan grassland" was interesting. However, there are some critical issues, which may need to be addressed.

[Comment] The statistical analysis and reporting are weak. Is there any real field replication, excluding any pseudo replication? What was the power of the statistical test? Statistical differences among different treatments were not reported for all the sub-plots. Additionally, along with p values, standard Error of the mean difference may need to be reported in the plots to understand the differences between the treatment means better.

[Responses] **Yes, we had real field replication, our site is a two-way factorial design (warming and grazing) was used with four replicates of each of four treatments. In total, 16 plots of 3-m diameter were fully randomized throughout the study site. We had shown it in our manuscript, please see the lines 144-148.**
**About the statistical differences among different treatments, it was also mentioned by other reviewer, as his suggestion, we removed the different letters from the Figs. 1b and 4e to avoid the misunderstandings. We also showed the two-way ANOVA results in Table 1 to give more details of statistical analysis in our manuscript. Please see lines 584-646.**

[Comment] It was not clear how were the relative contributions of bacteria and fungi in nitrification, denitrification and total $N_2O$ production derived from the total respective measurements? The methods need to be clear and reproducible.

[Responses] **For the contribution of bacteria and fungi to total nitrification enzyme activity was calculated it by the ratio of BNEA or FNEA to BNEA+FNEA; the contribution of bacteria and fungi to total potential of $N_2O$ production from denitrification was calculated it by the ratio of BDEA or FDEA to BDEA+FDEA. In the new version, we added the description in Materials and Methods". Please see the lines 256-259.**

[Comment] In addition to the present results of the relative contribution of bacteria and fungi in nitrification and denitrification, the definite mechanisms for bacterial and fungal pathways of nitrification and denitrification need to present to demonstrate the change in the pathway of $N_2O$ production under the warming treatment. A definite mechanism of shifting in the relative contribution of bacteria and fungi in $N_2O$ production would help the reader to understand the present results in a systematic way, particularly under the warming treatment. This would also help to explain and understand the underline reasons of changing the pathway of $N_2O$ production between bacteria and fungi under warming.

[Responses] **It is the two reasons that lead to the changes of fungal and bacterial pathways for $N_2O$ emissions by warming. Firstly, the increased of soil temperature**

directly reduce fungal activity but increase bacterial activity, because fungi prefer the cold environment compared with bacteria. Secondly, warming indirectly reduce fungal activity but increase bacterial activity through increased soil inorganic N and decreased soil organic N in our site, please see lines 352-363, because fungi prefer higher organic N environment while bacteria prefer higher inorganic N environment. All these changes caused the fungal and bacterial pathways for $N_2O$ emissions changed in different directions under warming. We have improved the manuscript and make sure the underlying mechanisms is clearly, please see lines 352-377.

[Comment] It was also not clear why the effects of warming on relative contribution of bacteria and fungi on nitrification, denitrification were diluted when warming treatment was combined with grazing, for example in fig 5?

[Responses] Yes, the effects of warming on relative contribution of bacteria and fungi on nitrification, denitrification were diluted when warming treatment was combined with grazing in our results. We had discussed in above that warming changed the pathway of $N_2O$ production potential mainly through alter the soil temperature and the soil inorganic and organic N content. In our results, (WG) also reduced the positive effect of (W) on the soil temperature (Fig. 1b), and showed the trend of reduced the negative effect of (W) on the TC, TN and $NO_3^-$ content although the statistical analysis were not significantly (Fig. 2), moreover, the soil dissolved organic nitrogen content was significantly diluted when warming treatment was combined with grazing (data not shown), so the effect of (WG) on soil temperature and the substrate concentration caused the effects of warming on relative contribution of bacteria and fungi on nitrification, denitrification were diluted when warming treatment was combined with grazing.

The paper presented an interesting topic, which focused on fungi regulating the responses of $N_2O$ production to warming and grazing treatments in Tibetan grassland. The authors report several new information, such as an increased bacterial enzyme activity and a decreased fungal enzyme activity of regulating $N_2O$ emissions under warming treatment. The findings have implications for well-understanding the responses of $N_2O$ emissions to the scenario of climate change and/or disturbance. However, there are sevel concerns need to be addressed.

[Comment] The description of experimental desigh is not clear, particularly, there is a confusing in introducing winter grazing treatment. What is the reason for the selection of winter grazing treatment in present study? Tibetan grassland is experienced to be covered by snow, frozen soils, and the grass should be withered in winter. In the same plots, the ecological effects of winter grazing should be interferenced by previous different grazing treatments (lines 153-156). How to avoid it?

**[Responses] Sorry, our previous description caused the misunderstanding by the referee. In the new version, we clarified why we used winter grazing. On the Qinghai-Tibet plateau, winter grazing is very commonly and alpine meadows are generally classified into two grazing seasons, i.e. warm season grazing from June to September and cold season grazing from October to May even the grassland was covered by snow (Cui et al., 2015). Winter pasture contributed about 40% of the grazed area in Qinghai–Tibet Plateau (Fan et al. 2010). The grazing treatments form 2006-2010 in the same experimental platform showed the effects of warming and grazing on ecosystem during the growing seasons (Luo et al. 2010; Hu et al. 2010; Wang et al. 2012), here is that shown after the summer grazing was replaced by winter grazing, does the alpine meadow grassland ecosystem was still affected by grazing (Zhu et al. 2015; Che et al. 2018).**

**We had improved the description of the winter grazing treatment and make it more clearly, please see lines 159-174.**

[Comment] Potential total nitrification/denitrification for $N_2O$ emission rate from incubation experiment is not a "real" rate of $N_2O$ emission under the field conditions. In terrestrial ecosystems, soil temperature, moisture, pH, soil N availability, and DOC etc. are generally considered as the major factors of controlling $N_2O$ emissions. For this study, the lack of field simultaneous monitoring data of $N_2O$ rates is a critical issue. Although the authors tried to cite the previous results for discussion, the conclusion obtained from an incubation experiment is still not general acceptable.

**[Responses]** We fully agree with the referee that the fungal and bacterial enzyme activities cannot be shown as the result of $N_2O$ emissions. The measurements under laboratory incubation reflected the potential ability of the soil fungal and bacterial activities in nitrification and denitrification because such laboratory incubation could avoid the impacts of various confounding factors and well clarify the mechanism responsible for $N_2O$ produce process.

For the lack of field simultaneous monitoring data of $N_2O$ rates, because our study was focused on the microbial mechanism responsible for $N_2O$ produce process but not for the $N_2O$ flux, so we think the field $N_2O$ emission is not necessary. There are also a series of studies showed the microbial mechanism responsible for $N_2O$ produce process and conclusions by incubation experiment, eg. Zhong et al. (2015, 2017); Huang et al. (2017); Marusenko et al. (2013); Attard et al. (2011) and so on.

At revised version, we clarified that our measurements in the laboratory indicated the potential emission to reveal the mechanism responsible for $N_2O$ produce process but not the field emission.

[Comments] The underlying mechanisms that fungal and bacterial pathways for controlling $N_2O$ emissions remain unknown. The authors need to elaborate the relative contributions of fungi and bacteria in nitrification and denitrification processes of $N_2O$ productions.

**[Responses]** It is the two reasons that lead to the changes of fungal and bacterial pathways for $N_2O$ emissions by warming. Firstly, the increased of soil temperature directly reduce fungal activity but increase bacterial activity, because fungi prefer the cold environment compared with bacteria. Secondly, warming indirectly reduce fungal activity but increase bacterial activity through increased soil inorganic N and decreased soil organic N in our site, please see lines 352-358, because fungi prefer higher organic N environment while bacteria prefer higher inorganic N environment. All these changes caused the contribution of fungi in nitrification and denitrification was reduced by warming, but the contribution of bacteria in nitrification and denitrification was increased by warming (Fig.5), then due to the fungal and bacterial pathways for $N_2O$ emissions was changed in different directions under warming.

We have improved the manuscript and make sure the underlying mechanisms is clearly, please see Lines 352-377.

[Comment] Line 130-131: The symbol oC is not correct.

**[Responses] Thank you for your suggestion. We had corrected it,please see lines 136-137.**

[Comment] There are several mistakes in English writing, which should be revised throughout the text.

**[Responses]** In the new version, we almost rewrote the manuscript and asked a native English speaker Miss Ri Weal to polish the language errors. We hope the new version is easy to read and follow.

**To M. W. I. Schmidt,**

[Comment] The methods used seem appropriate in general, however some questions arise with regards to measurements and sampling. The paper leaves open why the difference was set to 1.2_C and 1.7_C during day and night respectively in summer. Furthermore, it is not clear what the effect of 1500 W are in winter. The authors mention that some thermometers are broken, but it would have been nice to get at least the data from the working thermometers. Zhong et al. (2018) mention from the begin-ning and in the title, that the effect of winter grazing was under investigation. However, for half of the time there was summer grazing on the sites. Please describe this treatment further.

**[Responses] For "why the difference was set to 1.2 °C and 1.7 °C during day and night respectively in summer." Before we set up the field site, we have done an experiment to make sure the set of warming treatment can be succeeded used for stimulating climate warming in alpine meadow grassland, and proved the set of temperature was good. The FATE heating system was described by Kimball et al. (2008).**

**For "the effect of 1500 W are in winter", In summer, the power output of the heaters was manually set at 1500 W per plot was enough to increase the soil temperature as our treatment's set. But the temperature is very cold in winter, so some infrared thermometers were not working. To make sure the warming treatment was the same in summer and winter, the power output of the heaters was manually set at 1500 W per plot to make sure the increased of soil temperature were also 1.2 °C during the daytime and 1.7 °C at night in winter. We have improved it, please see lines 156-157.**

**For "half of the time there was summer grazing on the sites." Sorry, our previous description caused the misunderstanding by the referee. In the new version, we clarified why we used summer grazing and winter grazing. On the Qinghai-Tibet plateau, winter grazing is very commonly and alpine meadows are generally classified into two grazing seasons, i.e. warm season grazing from June to September and cold season grazing from October to May even the grassland was covered by snow (Cui et al., 2015). Winter pasture contributed about 40% of the grazed area in Qinghai–Tibet Plateau (Fan et al. 2010). The grazing treatments form 2006-2010 in the same experimental platform showed the effects of warming and grazing on ecosystem during the growing seasons (Luo et al. 2010; Hu et al. 2010; Wang et al. 2012), here is that shown after the summer grazing was replaced by winter grazing, does the alpine meadow grassland ecosystem was still affected by grazing (Zhu et al. 2015; Che et al. 2018). We had improved the**

**description of the winter grazing treatment and make it more clearly, please see lines 159-174.**

[Comments] General: Samples were taken on one only day. Would it be possible, that due to special environmental circumstances on that day, the results were in some way not representative?

**[Responses] In ecological studies of grassland, a series of studies only samples one time to show the effects of treatments on ecosystems (Zhong et al. 2015, 2017; Che et al. 2018; Marusenko et al. 2013). This is very commonly in ecological studies of grassland. And before we soil sampling, we also had checked the weather condition in the previous week, and make sure the weather condition of sampling time is commonly and the samples were representative.**

[Comments] The authors do not explain why they chose to simulate winter grazing and not summer grazing. If it is the reason mentioned in line 370, the authors should explain it already in the introduction. Consider using less acronyms, it is sometimes hard to follow the story. Rethink if 'treatment' (e.g. summer grazing treatment) really needs to be used that often. In the introduction: Maybe elaborate more on the state of art and on similar studies done in other parts of the world.

**[Responses] Sorry, our previous description caused the misunderstanding by the referee. In the new version, we clarified why we used winter grazing. On the Qinghai-Tibet plateau, winter grazing is very commonly and alpine meadows are generally classified into two grazing seasons, i.e. warm season grazing from June to September and cold season grazing from October to May (Cui et al., 2015). Winter pasture contributed about 40% of the grazed area in Qinghai–Tibet Plateau (Fan et al. 2010). The grazing treatments form 2006-2010 in the same experimental platform showed the effects of warming and grazing on ecosystem during the growing seasons (Luo et al. 2010; Hu et al. 2010; Wang et al. 2012), here is that shown after the summer grazing was replaced by winter grazing, does the alpine meadow grassland ecosystem was still affected by grazing (Zhu et al. 2015; Che et al. 2018). We had improved the description of the winter grazing treatment and make it more clearly, please see lines 66-174.**

[Comments] 259: What does (w/w) mean?

**[Responses] It is an abbreviation for "by weight," it is quite commonly used to describe the soil moisture.**

[Comments] 290: What does the sentence mean? There were no differences in the contribution of FNEA and FDEA to TNEA and TDEA in any treatments. There are differences, aren't there?

**[Responses] Yes, there are differences on the contribution of FNEA and FDEA to TNEA**

**and TDEA in treatments, we had removed this sentence to avoid the misunderstanding, please see lines 304.**

[Comments] 306: What does 'high complex compound substract substrate' mean?

**[Responses] It is means the organic matter, for easier understanding, we had changed it as high organic substrate, please see lines 319.**

[Comments] 336/369: Please elaborate on 'field data from 2011-2012'? It is not clear to us what this refers to.

**[Responses] The field data from 2011-2012 means the $N_2O$ emissions in the year of 2011-2012 in our site. We had improved these sentences, please see the lines 347-349 and 394.**

[Comments] 341: We do not understand the sentence starting with "In our site: : :". Maybe you can clarify /reformulate that.

**[Responses] We changed it as in our results to avoid the misunderstanding, please see lines 358.**

[Comments] 364f: We do not understand the sentence starting with "Additionally: : :". Please elaborate on why the effect of sheep is limited compared to other livestock?

**[Responses] "Additionally" means the other reasons. For "Please elaborate on why the effect of sheep is limited compared to other livestock?". Sorry, there is a mistake in this sentence, we have corrected it, please see lines 389-390.**

[Comments] Figure 1 and 4: The distribution of the letters indicating the significant differences is inconsistent, why do you only show it in section b of figure 1 and in section e of figure 4? Also think about using other symbols, since these letters might be confused with the letters for the figure subdivision.

**[Responses] We have removed different letters from the Figs. 1b and 4e. We also showed the two-way ANOVA results in Table 1 to give more details of statistical analysis in our manuscript. Please see lines 584-646.**

[Comments] Figure 5: From this figure we read that fungi and bacteria come from the hard rock substrate, that the denitrification happens in the subsoil and the nitrification in the topsoil. Is that right? Furthermore, we do not understand why W and WG are yellow shadowed and why 'bacteria' is written in purple, while the arrow is green. Also, it is not necessary to make the figure in 3D.

**[Responses] The contribution of fungi and bacteria to nitrification and denitirification all showed the results of topsoil, because the soil of this study was belong to topsoil and collected at a depth of 0–20 cm. We had improved the figure 5 to avoid the misunderstanding, please see figure 5.**

[Comments] Typos/ remarks concerning structure: 54f: 'Potential' is used too many times.

**[Responses] The word is necessary; it can avoid confusion with N₂O flux.**

[Comments] 153: This sentence is formulated rather complicated, maybe you can split it in two sentences. The term 'grazing treatments' is repeated a lot in those lines, maybe you can replace it?

**[Responses] We had improved it, please see lines 159-161.**

[Comments] 220: Denitrification enzyme activity

**[Responses] Corrected, please see lines 230.**

[Comments] 278: forgot N in unit

**[Responses] Corrected, please see lines 293.**

[Comments] 279: forgot N in unit

**[Responses]Corrected, please see lines 293.**

[Comments] 316-318: Does this conclusion not contradict to line 103? 322: nitrification and denitrification.

**[Responses] We had improved this sentence to avoid the misunderstanding, please see lines 325-329.**

[Comments] 334: fungal N₂O production potential.

**[Responses] Corrected, please see lines 345.**

[revised manuscript text omitted]

Fig. 2

[Figure]

Fig. 3

Fig.4

[Figure]

Fig.5

[Figure]